# KORGym: A Dynamic Game Platform for LLM Reasoning Evaluation

**Jiajun Shi**[1,2,3] *, **Jian Yang**[1,2] *†, **Jiaheng Liu**[2,4], **Xingyuan Bu**[2], **Jiangjie Chen**[3],
**Junting Zhou**[2], **Kaijing Ma**[2,3], **Zhoufutu Wen**[2,3], **Bingli Wang**[2], **Yancheng He**[2],
**Liang Song**[2], **Hualei Zhu**[2], **Shilong Li**[2], **Xingjian Wang**[2], **Wei Zhang**[2], **Ruibin Yuan**[2],
**Yifan Yao**[2], **Wenjun Yang**[2], **Yunli Wang**[2], **Siyuan Fang**[2], **Siyu Yuan**[4], **Qianyu He**[4],
**Xiangru Tang**[2], **Yingshui Tan**[2], **Wangchunshu Zhou**, **Zhaoxiang Zhang**[5], **Zhoujun Li**[1],
**Wenhao Huang**[3,†], **Ge Zhang**[2,3,†],
[1]SKLCCSE, Beihang University    [2]M-A-P    [3]ByteDance Seed    [4]Nanjing University [5]CASIA
* Equally contributed authors   †Corresponding author

## Abstract

Recent advancements in large language models (LLMs) underscore the need for more comprehensive evaluation methods to accurately assess their reasoning capabilities. Existing benchmarks are often domain-specific and thus cannot fully capture an LLM's general reasoning potential. To address this limitation, we introduce the **Knowledge Orthogonal Reasoning Gymnasium (KORGym)**[1], a dynamic evaluation platform inspired by KOR-Bench [1] and Gymnasium [2]. KORGym offers over fifty games in either textual or visual formats and supports interactive, multi-turn assessments with reinforcement learning scenarios. Using KORGym, we conduct extensive experiments on 19 LLMs and 8 VLMs, revealing consistent reasoning patterns within model families and demonstrating the superior performance of closed-source models. Further analysis examines the effects of modality, reasoning strategies, reinforcement learning techniques, and response length on model performance. We expect KORGym to become a valuable resource for advancing LLM reasoning research and developing evaluation methodologies suited to complex, interactive environments.

## 1 Introduction

Recent advances in reasoning models have yielded strong performance in tasks such as textual comprehension [3] and logical inference [4]. However, most benchmarks remain domain-specific (e.g., AIME [5], PHYBench [6]) and fail to capture general reasoning ability. Even benchmarks intended to evaluate broader reasoning (e.g., SuperGPQA [7], HLE [8]) are heavily influenced by pretraining data, limiting their capacity to measure intrinsic reasoning skills. To address this gap, we propose a benchmark designed to evaluate the intrinsic reasoning capabilities of LLMs independent of pretraining knowledge. Games, with their diverse scenarios rarely encountered in pretraining corpora, offer an ideal testbed for such evaluation.

While games offer a promising benchmark medium, existing approaches exhibit several shortcomings. LogicGame [9], for example, employs only single-turn scenarios, preventing evaluation of long-term planning in LLMs. TextArena [10] and SPINBench [11] support multi-turn scenarios but introduce opponent dynamics that generate extraneous variability, confounding pure reasoning assessment and limiting suitability for reinforcement learning (RL) by enabling hacked strategies. Moreover,

---

[1]Our codebase and experimental results are available at: `https://github.com/multimodal-art-projection/KORGym`

gg-bench [12] relies heavily on generative capacity and lacks robustness in both gameplay fidelity and RL integration.

To overcome these limitations, in Figure 1, we introduce the **K**nowledge **O**rthogonal **R**easoning **Gym**nasium (**KORGym**), inspired by the knowledge-orthogonal reasoning framework of KOR-Bench [1] (see Appendix C) and built on the reinforcement-learning environment Gymnasium [2].

Specifically, KORGym features over fifty games spanning six reasoning dimensions: mathematical and logical reasoning, control interaction reasoning, puzzle reasoning, spatial and geometric reasoning, strategic reasoning, and multimodal reasoning. The platform is organized into four modular components—the inference module, game interaction module, evaluation module, and communication module—enabling multi-round evaluations, configurable difficulty levels, and stable reinforcement-learning support.

By integrating textual and multimodal challenges, KORGym provides a comprehensive assessment of LLMs' adaptability, strategic planning, and decision-making capabilities, thereby offering a more accurate reflection of their intrinsic reasoning abilities. Using KORGym, we conduct extensive experiments, yielding several key insights:

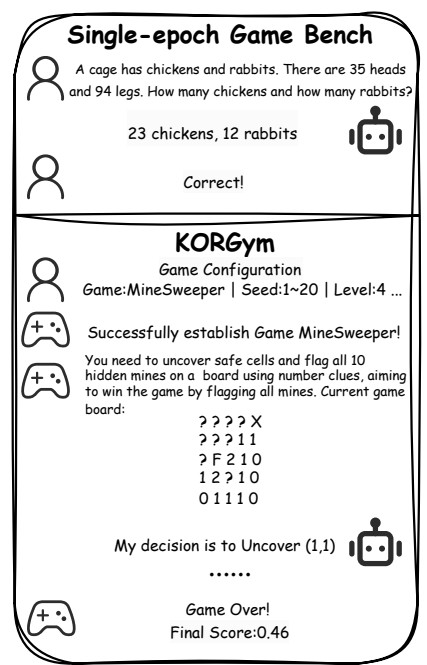

- Reasoning abilities display consistent profiles of strengths and weaknesses within the same model series.

- Modality influences reasoning performance, with distinct patterns observed between open-source and closed-source LLMs.

- Thinking models exhibit behavioral patterns distinct from those of non-thinking models.

Figure 1: Comparison Between Traditional Single-epoch Game Benchmark and KOR-Gym.

- LLMs often employ explicit reasoning paradigms during problem-solving, which may partially constrain their performance.

- Appropriate reinforcement learning enhances reasoning capabilities and yields more balanced performance across different reasoning dimensions.

In summary, our main contributions are as follows:

- We design a suite of **over fifty text- and vision-based games** tailored to evaluate the reasoning capabilities of large language models.

- We present **KORGym**, an extensible framework supporting incremental development and reinforcement-learning integration.

- We conduct a comprehensive empirical analysis of 19 LLMs and 8 vision-language models and uncover several key insights.

## 2 Related Work

**LLMs for Gaming.**  Games serve as valuable testbeds for evaluating large language models (LLMs) due to their demands for multi-step reasoning and strategic planning. Early research focused on single-game evaluations in domains like Minecraft [13] or social deduction games [14, 15], but these narrow settings limited generalizability. Subsequent efforts introduced broader benchmarks with diverse game types and multi-agent frameworks emphasizing coordination or competition, though critical dimensions such as open-ended negotiation, dynamic cooperation-conflict shifts, and rich social dynamics remained underexplored. Some researchers have designed PlanBench [16] to evaluate the long-term reasoning ability of LLMs and have further conducted studies [17] on reasoning models

based on this work. Meanwhile, other researchers have proposed the Game Traversal Benchmark [18] to assess LLMs' planning and reasoning ability through the task of traversing 2D game maps. To address these gaps, SPIN-Bench [11] unifies strategic planning and social intelligence by combining formal planning analysis, multi-agent cooperation/competition, and open-ended dialogue. Existing benchmarks vary widely in environment diversity and technical capabilities. Some frameworks offer diverse environments but lack human evaluation, while others focus on specific scenarios yet miss key features. SPIN-Bench stands out with a balanced mix of game types, Gym compatibility, and model vs. model evaluation.

**Knowledge Orthogonality Based Evaluation.** Current AI reasoning benchmarks (e.g., MMLU [19], CommonsenseQA [20], MATH [21]) emphasize factual recall and problem-solving but often conflate memorization with reasoning, limiting insight into underlying cognitive processes. To address this, integration-based benchmarks (e.g., ZebraLogic [22], TravelPlanner [23]) test adapt-ability and creativity by requiring pattern recognition, logic, and multi-step reasoning in novel contexts. While these frameworks advance the focus on contextual problem-solving, they still risk entanglement with domain-specific knowledge biases, as seen in mathematical or logical benchmarks like GSM8K [24] and FOLIO [25]. To address these gaps, the concept of knowledge orthogonality advocates decoupling reasoning assessment from prior knowledge and prioritizing rule-following in out-of-distribution scenarios to isolate core abilities such as systematic generalization and hypothesis testing. This paradigm shift—from memorization-driven metrics to knowledge-agnostic, creativity-focused evaluations—establishes a fairer framework for measuring cognitive agility, ensuring models demonstrate genuine reasoning rather than reciting learned patterns and fostering AI systems with robust, human-like adaptability in open-world environments.

# 3 Approach

## 3.1 Framework

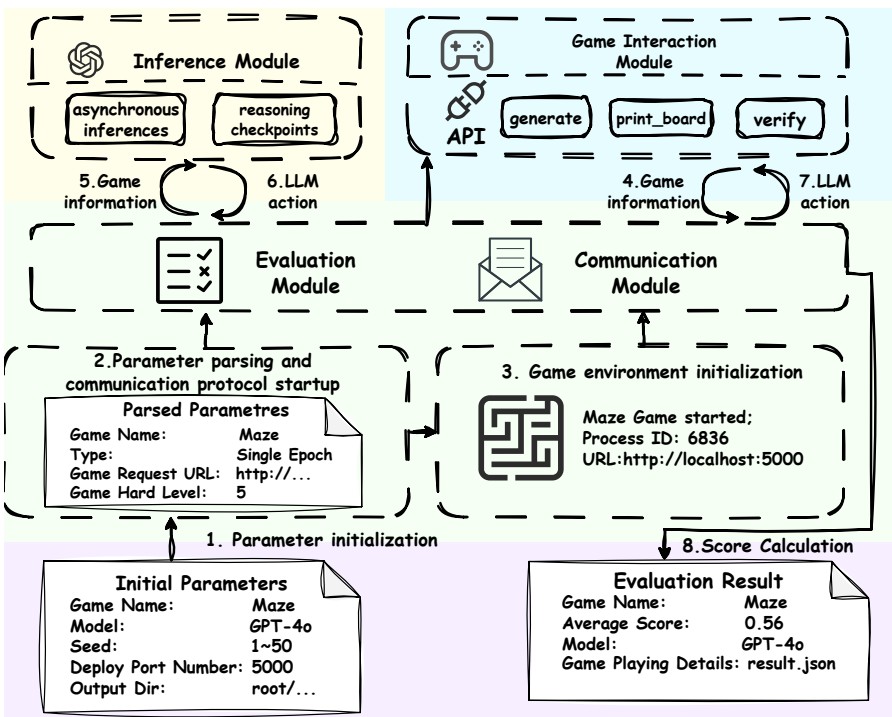

Figure 2: Framework of the KORGym system. Our system architecture primarily consists of four modules: the Inference Module, Game Interaction Module, Evaluation Module, and Communication Module. The initialization parameters include: Game Name, Model Information, Seed, Deployment Port Number, and Output Directory.

We propose **KORGym**, an efficient game-based framework for evaluating the complex reasoning capabilities of large language models (LLMs) via both single-turn and multi-turn text-based and multimodal games. KORGym is organized into three key modules:

- **Evaluation and Communication Module**: the system core, which parses input parameters, establishes inter-module communication protocols, encapsulates and transmits communication packets, and logs final evaluation scores.
- **Game Interaction Module**: encapsulates the game environment and interaction APIs, including:
  - **generate**: initializes the game environment.
  - **print board**: renders the game board and generates prompts.
  - **verify**: updates the game state and computes scores.
- **Inference Module**: manages model inference processes, including asynchronous acceleration and intermediate result checkpointing.

Based on these modules, the primary inference workflow of KORGym proceeds as follows:

- **Parameter Initialization**: Load initial parameters (e.g., game type, seed, and model).
- **Parameter Parsing and Communication Protocol Startup**: Parse these parameters and encapsulate them into communication packets.
- **Game Environment Initialization**: Generate the game environment according to the parameters.
- **Acquisition of Game Information**: Invoke the generate and print board APIs in the Game Interaction Module to obtain the current environment state.
- **Game Information Transmission**: Package the retrieved information for transmission.
- **Generation of LLM Action**: Perform model inference to generate the next action.
- **Transmission of LLM Action**: Send the action via the verify API to update the game state; if the game is not concluded, return to Acquisition of Game Information.
- **Score Calculation and Result Output**: Compute and output the final score.

## 3.2 Task Introduction

As illustrated in Figure 9 (Appendix H), KORGym supports over fifty novel games, enabling precise and efficient evaluation of the reasoning abilities of large language models (LLMs) across six distinct dimensions (Table 11): **Mathematical and Logical Reasoning**, **Control Interaction Reasoning**, **Puzzle Reasoning**, **Spatial and Geometric Reasoning**, **Strategic Reasoning** and **Multimodal Reasoning**.

During benchmark development, we selected more than fifty games that effectively capture the reasoning capabilities of LLMs. These games span four categories: traditional puzzles (e.g., Sudoku); adaptations of classic video games (e.g., Plants vs. Zombies; Minesweeper); game-theoretic challenges (e.g., N-point; Evolution of Trust); and multimodal tasks (e.g., Jigsaw; Circle the Cat).

KORGym offers a suite of over fifty games—with continuous expansion—that support multi-turn interactions via standardized APIs (generate, verify, and print board). The platform is tailored for RL, providing environment states and reward signals, and enables users to adjust game difficulty and environmental diversity through scalable parameters. Additionally, it includes nine multimodal games, facilitating comprehensive evaluation in both textual and multimodal contexts. Related platforms include LogicGame [9], AgentBench [26], GameArena [27], SPIN-Bench [28], TEXTARENA [10], and ReasoningGYM [29]. A detailed comparison appears in Table 10 of Appendix I.

## 3.3 Evaluation Method

**Score Calculation Rules**    To address the limitations of binary (0/1) scoring in reflecting intermediate progress in KORGym, we propose a comprehensive scoring scheme comprising three rules:

- **Binary Scoring**: For single-objective games, assign 1 point for success and 0 for failure. For example, in 7-Maze, reaching the exit yields a score of 1.

- **Proportional Scoring**: For multiple-choice games, the score equals the number of correct responses divided by the total number of options. For instance, in the 44-Jigsaw Puzzle, the score is the number of correctly placed pieces over the total pieces.
- **Cumulative Scoring**: For games that award incremental points, accumulate all points earned. For example, in 3-2048, each tile merge contributes to the final score.

**Capability Dimension Aggregated Mean**   Raw game scores in KORGym can extend beyond the $[0,1]$ interval and may be skewed by variations in game difficulty or by outlier model behaviors. To mitigate these issues, we introduce **Capability Dimension Aggregated Mean**, a more robust aggregation metric for evaluating model performance across reasoning dimensions.

Formally, let $G = \{g_1, g_2, \ldots, g_N\}$ denote the set of all games, $M = \{m_1, m_2, \ldots, m_K\}$ denote the set of models under evaluation, and $D = \{d_1, d_2, \ldots, d_L\}$ represent the set of reasoning capability dimensions. Each game $g \in G$ is associated with a specific dimension $d(g) \in D$. Let $S_{g,m}$ denote the raw score achieved by model $m \in M$ on game $g \in G$. For each game $g$, if the maximum score across all models exceeds 1, i.e., $\max_{m \in M} S_{g,m} > 1$, we apply a $\log p$ transformation (i.e., $\ln(1 + x)$) to compress large score values and reduce skewness; otherwise, we retain the original score:

$$S'_{g,m} = \begin{cases} \ln\big(1 + S_{g,m}\big), & \text{if } \max_{m \in M} S_{g,m} > 1, \\ S_{g,m}, & \text{otherwise.} \end{cases} \tag{1}$$

To normalize scores across games, we further define, for each game $g$,

$$a_g = \min_{m \in M} S'_{g,m}, \quad b_g = \max_{m \in M} S'_{g,m}. \tag{2}$$

If $b_g = a_g$, meaning all models perform identically on game $g$, we assign every model a normalized score of 0.5 to avoid division by zero. Otherwise, we normalize its adjusted score:

$$\widetilde{S}_{g,m} = \frac{S'_{g,m} - a_g}{b_g - a_g}, \quad \forall m \in M. \tag{3}$$

This normalization ensures that for each game, model performances are mapped into the $[0, 1]$ range while preserving relative differences. Subsequently, for each capability dimension $d \in D$, we define the corresponding set of games and aggregated score of model $m$ on dimension $d$ as:

$$G_d = \{g \in G : d(g) = d\}, \overline{S}_{d,m} = \frac{1}{|G_d|} \sum_{g \in G_d} \widetilde{S}_{g,m}. \tag{4}$$

The resulting matrix $\{\overline{S}_{d,m}\}_{d \in D, \, m \in M}$ provides a normalized, dimension-wise evaluation of reasoning capabilities that is fair across heterogeneous games.

## 4   Experiments

### 4.1   Settings

**LLMs**   To comprehensively evaluate LLM performance, we assessed 19 large language models—including 11 thinking models and 8 instruction-tuned models—and 8 vision-language models (Table 12).

**Evaluation Setting**   During evaluation, we apply distinct protocols for single-epoch and multiple-epoch games:

- Single-epoch Games: Each model is evaluated on 50 independently initialized game instances by varying the "seed" parameter in the "generate" API from 1 to 50.
- Multiple-epoch Games: For each model, we initialize 20 game environments. Each episode permits up to 100 interaction rounds, and we vary the "seed" parameter in the "generate" API from 1 to 50 for reproducibility.

All assessments use a zero-shot prompting setup to gauge genuine reasoning capabilities, retaining each model's default sampling parameters (temperature and top-p). We evaluate closed-source models via their hosted APIs and open-source models on eight NVIDIA A100-80G GPUs.

Table 1: Overall performances of different models on KORGym. Model capability dimensions include Mathematical and Logical Reasoning (MLR), Control Interaction Reasoning (CIR), Puzzle Reasoning (PR), Spatial and Geometric Reasoning (SGR) and Strategic Reasoning(SR).

| Model | MLR(%) | CIR(%) | PR(%) | SGR(%) | SR(%) | Avg.(%) |
|---|---|---|---|---|---|---|
| O3-mini | 77 | 81 | 79 | 94 | 76 | 82 |
| Gemini-2.5-pro-03-25 | 63 | 94 | 93 | 59 | 84 | 79 |
| O1-2024-12-17 | 74 | 83 | 65 | 79 | 66 | 73 |
| Doubao-1-5-thinking-pro | 65 | 74 | 84 | 72 | 65 | 72 |
| DeepSeek-R1 | 66 | 82 | 69 | 56 | 83 | 71 |
| Claude-3.7-thinking | 50 | 93 | 52 | 53 | 64 | 62 |
| Qwen3-32B-thinking | 58 | 55 | 58 | 55 | 71 | 60 |
| DeepSeek-v3-0324 | 35 | 55 | 27 | 26 | 69 | 42 |
| DeepSeek-R1-Distill-Qwen-32B | 45 | 28 | 35 | 33 | 56 | 39 |
| Gemini-2.0-Flash-thinking | 25 | 53 | 34 | 18 | 58 | 38 |
| Claude-3.7 | 25 | 55 | 26 | 17 | 50 | 35 |
| Qwen-QwQ | 37 | 39 | 14 | 18 | 33 | 28 |
| Gemini-2.0-Flash | 24 | 28 | 17 | 12 | 51 | 26 |
| GPT-4o | 12 | 25 | 8 | 11 | 53 | 22 |
| Doubao-1.5-pro | 18 | 16 | 16 | 7 | 44 | 20 |
| Qwen2.5-72B-Instruct | 18 | 10 | 4 | 7 | 49 | 18 |
| Qwen2.5-32B-Instruct | 13 | 7 | 4 | 9 | 46 | 16 |
| DeepSeek-R1-Distill-Qwen-7B | 10 | 2 | 6 | 3 | 33 | 11 |
| Qwen2.5-7B-Instruct | 7 | 1 | 1 | 1 | 29 | 8 |

Table 2: Multimodal reasoning abilities of different models on KORGym.

| Model | Crossword Puzzle(%) | Jigsaw Puzzle(%) | Find The Pattern(%) | Circle The Cat(%) | Map Simulation (%) | Sokoban (%) | Bubble Ball Sorting(%) | Wordle (%) | Square Addition(%) |
|---|---|---|---|---|---|---|---|---|---|
| Doubao-vision-250115 | 14.4 | 12.9 | 42 | 0 | 2 | 4 | 0 | 0 | 0 |
| Gemini-2.5-Pro | 18.4 | 19.5 | 66 | 15 | 26 | 10 | 90 | 85 | 2 |
| Gemini-2.0-Flash | 24.9 | 12.7 | 54 | 5 | 0 | 4 | 45 | 15 | 0 |
| GPT-4o | 23.7 | 8.4 | 36 | 0 | 0 | 4 | 65 | 15 | 0 |
| Qwen2.5VL-72B | 14.4 | 10.9 | 28 | 0 | 2 | 4 | 20 | 5 | 0 |
| Qwen2.5VL-32B | 6.4 | 8.6 | 28 | 0 | 0 | 2 | 25 | 5 | 0 |
| Qwen2.5VL-7B | 4.4 | 0 | 16 | 0 | 0 | 0 | 0 | 0 | 0 |
| InternVL3-78B | 11.8 | 10 | 38 | 0 | 2 | 0 | 25 | 0 | 0 |

## 4.2 Main Results

Table 1 reports the performance of LLMs and VLMs on KORGym using the Capability Dimension Aggregated Mean (Section 4.1). Table 2 presents VLM performance on multimodal tasks and detailed raw scores appear in Appendix D. The leaderboard will be updated on GitHub after submission. Across 51 games and six reasoning dimensions, the normalized scores yield the following insights.

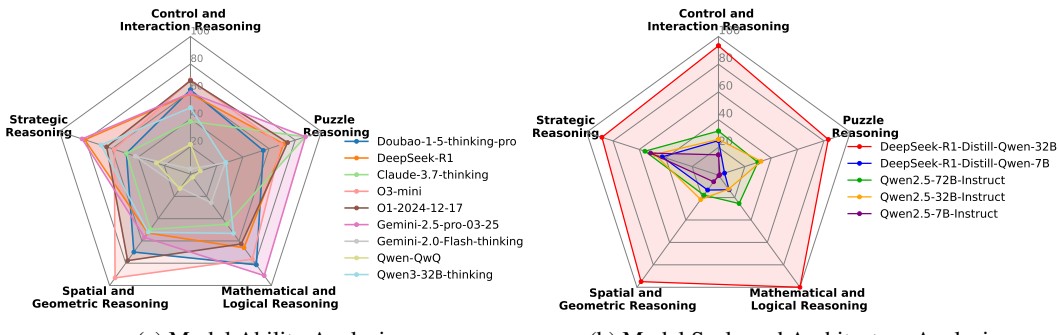

(a) Model Ability Analysis      (b) Model Scale and Architecture Analysis

Figure 3: Capability Dimension Illustration. Figure (a) showcases the performance of the top-performing models on KORGym. Figure (b) showcases the impact of model scale and architecture on reasoning capabilities.

**Similar Strength–Weakness Profiles Within Same Model Series**    Figure 3a shows that O1 and O3-mini excel in spatial reasoning, whereas the Gemini series leads in mathematical and puzzle reasoning.

**Closed-Source Models Demonstrate Superior Reasoning Performance**    O3-mini achieves the highest overall score on KORGym, particularly in spatial reasoning. Claude-3.7-thinking and Gemini-2.5-pro top puzzle reasoning, while Doubao-1.5-thinking-pro and DeepSeek-R1 deliver balanced performance across dimensions. In contrast, open-source models lag behind.

**Impact of Model Scale and Architecture on Reasoning Capabilities**    Figure 3b demonstrates that model performance scales positively with model size and thinking models outperform size-matched non-thinking variants. For instance, DeepSeek-R1-Distill-Qwen-32B, though smaller in scale, exceeds the performance of Qwen2.5-72B-Instruct.

## 5    Discussion

### 5.1    RQ1: Does Modality Affect Reasoning Performance?

**Textual-version Game vs. Multimodal-version Game.**    As shown in Figure Figure 4 compares the performance of closed-source and open-source VLLMs on textual and visual versions of six representative games. Detailed results are provided in Figure 10 of Appendix J. Our key findings are:

- **Average scores on textual versions consistently exceed those on visual versions.**

- **Open-source VLMs perform better on text-based reasoning than on visual-based tasks**, indicating limited visual grounding or underdeveloped multimodal alignment.

- **Some closed-source VLMs score higher on visual versions than on textual versions**, suggesting stronger visual reasoning or superior multimodal integration.

- **In mathematics-related games, models score significantly higher on textual versions,** highlighting the advantage of symbolic representation for numerical reasoning.

### 5.2    RQ2: Do Different Model Series Exhibit Consistent Behavioral Patterns?

Based on the experimental scores, we computed the mean and standard deviation of each dimension's scores (Figure 5a) and performed principal component analysis (PCA) on the score matrix $\hat{S} \in R^{M \times G}$, where $M$ and $G$ denote the numbers of models and games, respectively (Figure 5b). These analyses reveal dominant patterns in the models' reasoning behavior across five dimensions.

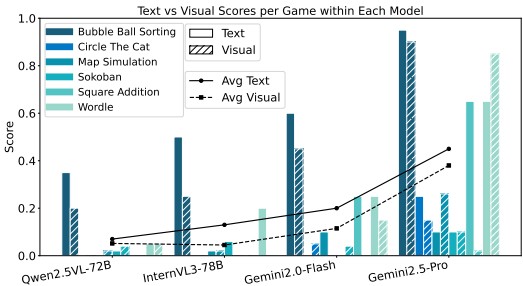

**Top-Tier Models Exhibit Homogeneous Behavioral Profiles**    In PCA space, top-tier models (e.g., O1 and O3-mini) form tight clusters, indicating consistently strong reasoning performance across all dimensions. By contrast, secondary models (e.g., Claude-3.7-thinking and Qwen3) display imbalanced performance across reasoning dimensions.

Figure 4: Performance Comparison Between Textual and Multimodal Game Versions. This figure illustrates a given model's performance on both the textual and multimodal versions of the same game. Different games are represented by distinct bar colors, and bar shading differentiates text (unshaded) from visual (shaded) versions. Solid and dashed lines correspond to the average textual and visual scores, respectively.

**Distinct Behavioral Patterns Between Thinking and Non-Thinking Models**    PCA reveals that the first two clusters consist exclusively of thinking models, whereas the fourth cluster comprises almost solely non-thinking models.

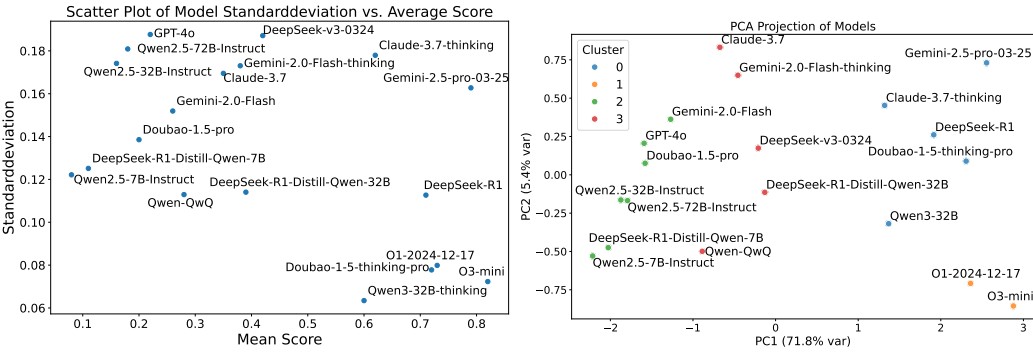

(a) Stability Analysis of Model Reasoning Capabilities      (b) PCA Projection of Models

Figure 5: Analysis of Model Behavioral Characteristics. (a) illustrates the relationship between model stability and reasoning performance, whereas (b) depicts the clustering of models based on behavioral traits.

**LLMs tend to adopt explicit reasoning paradigms when performing analysis and problem-solving** Response-level case studies reveal that LLMs in KORGym employ four primary reasoning paradigms, each reflecting a distinct cognitive strategy:

- **Code Paradigm**: generating executable code to obtain solutions (e.g., "import math; a = 0; for i in range(...): ...").
- **Mathematical Paradigm**: applying algebraic equations or arithmetic rules to model and solve problems (e.g., "Let x be the number of creature A and y the number of creature B; construct the system of equations...").
- **Algorithm-Specific Paradigm**: invoking established algorithms (e.g., Dijkstra's algorithm, Eulerian path) and adapting them to the task context (e.g., "Use an Eulerian path to solve the one-stroke drawing puzzle: first compute ...").
- **Natural Language Reasoning Paradigm**: conducting spatial, logical, or causal analysis in natural language (e.g., "If we turn right, we reach (1,2) where a springboard lies ahead ...").

To examine reasoning-paradigm usage, we employ GPT-4o to annotate model responses for selected KORGym games and compute the mean score for each paradigm (Table 3). We then conduct ablation experiments by constraining prompts to disable individual paradigms, with results summarized in Figure 6, detailed information in Appendix F. Our key findings include:

Table 3: Reasoning Paradigm Proportions and Average Scores for Different Models' Responses

| Model | Proportion (%) | | | | Average Score(%) | | | | Overall Score(%) |
|---|---|---|---|---|---|---|---|---|---|
| | Code | Math | Algorithm | Natural Language | Code | Mathematical | Algorithm | Natural Language | |
| Doubao-1-5-thinking-pro | 0.6 | 14 | 32.9 | 52.6 | 50 | 82 | 0.61 | 66 | 66.565 |
| Doubao-1.5-pro | 17.1 | 10.3 | 42.6 | 30 | 0 | 0 | 6 | 6 | 4.356 |
| DeepSeek-v3-0324 | 0 | 0 | 9.7 | 90.3 | 0 | 0 | 26 | 22 | 22.388 |
| Claude-3.7-thinking | 0.3 | 13.4 | 26.3 | 60 | 0 | 64 | 37 | 46 | 45.907 |
| Gemini-2.5-pro-03-25 | 46 | 6.9 | 12 | 35.1 | 50 | 83 | 100 | 66 | 63.893 |
| GPT-4o | 2 | 1.1 | 29.7 | 67.1 | 0 | 0 | 2 | 2 | 1.936 |
| O3-mini | 0 | 21.4 | 11.4 | 67.1 | 0 | 85 | 97 | 72 | 77.56 |

**Models within the same series exhibit distinct reasoning-paradigm preferences** Models tend to adopt paradigms aligned with their architecture: Gemini-2.5-Pro predominantly employs code-based reasoning; O3-mini primarily utilizes mathematical and natural language reasoning; and Doubao-1.5-thinking-pro strongly favors algorithm-specific reasoning strategies.

**Reasoning Paradigms Partially Constrain Model Performance** Disabling specific reasoning paradigms via prompt constraints led to in-

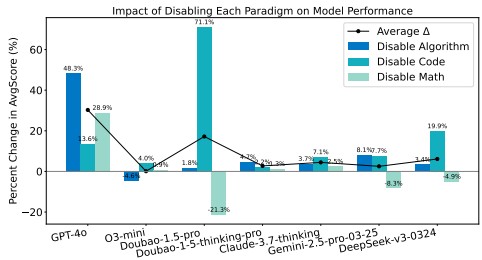

Figure 6: Impact of Disabling Each Paradigm on Model Performance

creased average performance across all models.

We attribute this improvement to overreliance on large-scale pretraining data rich in mathematical, coding, and algorithmic examples, which can impede generalization and adaptability to novel reasoning tasks. These findings underscore KORGym's value as a robust benchmark for evaluating genuine reasoning abilities beyond memorized patterns.

**Mathematical Reasoning as a Core Component of the Reasoning Process**    Disabling the mathematical paradigm causes most models to experience a performance decline or exhibit no improvement, indicating that mathematical reasoning is critical to LLM reasoning capabilities.

**Stronger Models Exhibit Greater Robustness**    More capable models are less impacted by disabling individual reasoning paradigms. For example, O3-mini and Doubao-1.5-thinking-pro maintain near-original performance when deprived of their preferred reasoning strategies, demonstrating superior robustness, generalization, and adaptive reasoning under constraint.

### 5.3   RQ3: What is the Impact of Reinforcement Learning (RL) on Problem Solving Capabilities?

During multi-turn reinforcement-learning fine-tuning, Doubao-1.5-thinking-pro incorporated two specialized algorithmic frameworks—DAPO and VAPO—to address instability in reasoning-oriented model training. In parallel, it was trained on a comprehensive corpus of **STEM problems, code-related tasks, logical reasoning challenges, and non-reasoning examples**. Additionally, RL training on classic games (e.g., 24-point, mazes, and Sudoku) yielded a marked improvement in its reasoning performance.

In KORGym, RL-driven enhancements yielded substantial gains across reasoning dimensions. As shown in Table 1, Doubao-1.5-thinking-pro achieved a mean score of 0.72—fourth overall—and excelled in puzzle reasoning with a score of 0.84, surpassing both O1 and O3-mini. Notably, Doubao-1.5-thinking-pro exhibits minimal performance degradation under ablation and demonstrates score variance comparable to leading models (e.g., O3-mini and O1-2024-12-17). These improvements underscore that appropriate reinforcement learning fosters both enhanced reasoning and more balanced performance.

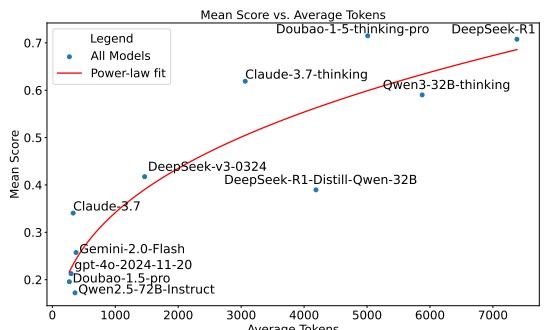

Figure 7: Correlation Between Response Length and Reasoning Performance

### 5.4   RQ4: Is There a Correlation Between Response Length and Reasoning Performance?

To examine the effect of response length on reasoning performance, we record token counts and reasoning scores for four reasoning models and eight non-reasoning models during gameplay. We then fit curves to the aggregated data to identify trends and correlations, as illustrated in Figure 7.

From the figure, we derive the following insights:

- **A strong positive correlation exists between reasoning performance and response length**: models with longer responses tend to achieve higher KORGym scores.

- **Reasoning and non-reasoning models differ markedly in response length distributions**: non-reasoning models produce responses within a narrow range, whereas reasoning models exhibit a broader and more varied distribution.

- **The impact of response length on performance exhibits diminishing returns**: as response length increases, incremental score gains become marginal, suggesting an upper limit to the benefits of verbosity.

### 5.5 RQ5: Does KORGYM contain data leakage?

Many benchmarks inevitably overlap with model training data. To show that KORGYM effectively reduces the extent of data leakage, we propose the Knowledge Impact Coefficient to quantitatively assess the degree of leakage. For a game within KORGYM, the required reasoning information comprises:

$K$: General background knowledge acquired during training.

$T$: Single-turn or multi-turn tasks that must be completed during gameplay.

$R$: Core game-rule information specifically designed for solving task $T$.

$A$: The set of actions generated by the LLM to solve task $T$.

$\rightarrow$: The reasoning and interaction process from the game task $T$ to action set $A$.

$S$: The final score obtained by the LLM's interaction with the game.

The leakage between $K$ and $R$ is quantified by the *Knowledge Impact Coefficient* $\beta$:

$$\beta = 1 - \frac{P(T \rightarrow A \mid R, K) - P(T \rightarrow A \mid K)}{P(T \rightarrow A \mid R, K)} = \frac{P(T \rightarrow A \mid K)}{P(T \rightarrow A \mid R, K)}, \quad \beta \in [0, 1].$$

A smaller $\beta$ indicates less overlap between the LLM's pre-trained knowledge and the game rules; consequently, the LLM cannot achieve optimal scores without explicitly relying on the rules provided in the prompt. Conversely, a larger $\beta$ indicates significant overlap, allowing the LLM to complete the game task and achieve optimal scores even without explicit rules.

| Model | Snake | Pipe Game | Long Cat | Sokoban | Word Transf. | 8-puzzle | Play Lines | Black–White Copy |
|---|---|---|---|---|---|---|---|---|
| Doubao-1-5-thinking-pro | 0.055 | 0.085 | 0 | 0.904 | 0 | 0.176 | 0 | 0.166 |
| DeepSeek-R1 | 0.333 | 0.200 | 0.208 | 0.636 | 0 | 0.307 | 0 | 0.133 |
| Qwen3-32B-thinking | 0.071 | 0.080 | 0 | 0.909 | 0 | 0.321 | 0 | 0.117 |
| o3-mini | 0.956 | 0.000 | 0 | 0.948 | 0 | 0.617 | 0 | 0.166 |
| Claude-3.7-thinking | 0.441 | 0.200 | 0.312 | 0.461 | 0 | 0.521 | 0 | 0.222 |

Table 4: Knowledge Impact Coefficient $\beta$ across selected KORGYM games.

Based on the above definition, we select a subset of games from KORGYM, including classic games (*Snake*, *Sokoban*, and *8-puzzle*) and original/adapted games (*Pipe Game*, *Long Cat*, . . . ). We conduct an ablation study in which the core game-rule information ($R$) is removed, and then compute the Knowledge Impact Coefficient $\beta$. The results are shown in Table 4.

- **Model perspective:** Models such as `o3-mini` and `DeepSeek-R1` exhibit relatively notable knowledge leakage, whereas others (e.g., `Doubao-1-5-thinking-pro`) demonstrate comparatively lower leakage. This observation indirectly supports the view that reinforcement learning (RL) processes genuinely enhance a model's reasoning capabilities, rather than merely enabling it to memorize training questions or regurgitate pre-training knowledge.

- **Game perspective:** Knowledge leakage is more prominent in classic games, while the corresponding $\beta$ values for original/adapted games remain relatively low. Given that more than $80\%$ of the games in KORGYM are originally developed or adapted by our engineering team, this strongly supports the validity and reliability of KORGYM.

## 6 Conclusion

In this paper, we introduce KORGym—a scalable, game-driven benchmark comprising over fifty tasks spanning six reasoning dimensions. KORGym supports multimodal interactions, reinforcement learning, and parameterized environments, and employs a robust evaluation methodology based on dimension-aware score aggregation tailored to game-based reasoning. We evaluate 19 LLMs and 8 VLMs, revealing consistent strength–weakness profiles within model series and demonstrating the impact of model scale and architecture on reasoning capabilities. We also conduct ablation studies on modality, model series, and reinforcement learning, providing a detailed analysis of the key factors influencing LLM reasoning performance.

## Acknowledgements

This work is supported by State Key Laboratory of Complex & Critical Software Environment (SKLCCSE). This work is also supported by the Fundamental Research Funds for the Central Universities.

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

# A   Limitations

While KORGym already offers a diverse suite of challenges, we recognize that its current catalog may not yet fully span the breadth of reasoning styles, difficulty levels, and domain contexts that real-world applications demand.

In particular, certain complex interaction patterns—such as multi-agent negotiation, long-horizon strategic planning, and fine-grained commonsense inference—are underrepresented at present. Moreover, the balance between textual and visual tasks could be further refined to ensure equitable coverage of each modality

To address these points, we intend to iteratively broaden KORGym's game portfolio—adding new thematic categories, adversarial and cooperative modes, and tasks designed to probe underexplored reasoning facets—while continuously revisiting our evaluation metrics to capture deeper, more nuanced model behaviors.

# B   Potential societal impacts

## B.1   Positive impacts

KORGym provides a rigorous, diverse, and extensible platform for evaluating the reasoning capabilities of large language models (LLMs), which can accelerate the development of safer, more generalizable, and more reliable AI systems. By analyzing LLM behaviors across varied reasoning tasks, KORGym contributes to improving the transparency, robustness, and alignment of such models—positively impacting applications in education, scientific discovery, and human-AI collaboration.

## B.2   Negative impacts

However, there are potential negative impacts as well. Enhanced reasoning capabilities in LLMs—if misused—could be exploited for harmful purposes, such as reinforcing algorithmic bias in high-stakes decision-making. To mitigate such risks, we emphasize the importance of responsible model deployment and include detailed evaluation metrics that can inform future alignment and safety research.

# C   Formal Definition of "Knowledge Orthogonality"

**For a task $T$, the required reasoning information consists of:**

- $K$: General background/domain-specific knowledge acquired during pre-training, excluding common sense.
- $R$: Core rule information designed to solve $T$.
- $Q$: A Rule-Driven question.
- $A$: Answer to the question $Q$.

**Notational Definitions:**

- $\rightarrow$: Represents the cognitive process of deriving $A$ from $Q$.
- $P$: Represents the belief strength that $A$ is a valid answer to $Q$ based on $R$ and/or $K$.
  - $P(Q \rightarrow A \mid R)$: Belief in $A$ driven solely by $R$.
  - $P(Q \rightarrow A \mid K)$: Belief in $A$ based solely on $K$.
  - $P(Q \rightarrow A \mid R, K)$: Combined belief in $A$, integrating $R$ and $K$.

**$T$ satisfies knowledge orthogonality under the following conditions:**

1. **Knowledge-Rule Decoupling**: Rule $R$ is logically self-contained and independent of $K$.

$$R \perp K$$

2. **Knowledge Assistiveness**: Background knowledge $K$ may support or interfere with the derivation of $A$ from $Q$, but does not play a central role in reasoning. The extent of this influence is quantified by the Knowledge Impact Factor ($\beta$), defined as:

$$\beta = \frac{P(Q \to A \mid R, K) - P(Q \to A \mid R)}{P(Q \to A \mid R)}$$

$\beta$ ranges from $(-1, \epsilon]$, where $\epsilon$ is a very small positive number.

- When $\beta$ is positive and close to 0, $K$ has little impact, with $R$ being dominant.
- When $\beta$ is negative, it can range from small negative values to approaching $-1$, where $K$ increasingly undermines reasoning.

3. **Rule Centrality**: Correctness relies on understanding and applying $R$, with $R$ having significantly greater influence than $K$.

$$P(Q \to A \mid R, K) \approx P(Q \to A \mid R) \gg P(Q \to A \mid K)$$

4. **Derivation Adjustment**: This formula adjusts the reasoning process based on $R$, incorporating the influence of $K$ with $\beta$ reflecting its effect.

$$P(Q \to A \mid R, K) = P(Q \to A \mid R) \cdot (1 + \beta)$$

# D Detailed Scores on KORGym

Table 5: Mathematical and logical reasoning abilities of different models on KORGym.

| Model | Date Calculation | Sudoku | Light Out Game | Square Addition | Alien | Party Time | Path Planning Problem | Construction Company | One Stroke Drawing | Coloring Issue | City Traveller |
|---|---|---|---|---|---|---|---|---|---|---|---|
| Doubao-1.5-pro | 0.060 | 0.120 | 0.060 | 0.000 | 0.200 | 0.240 | 0.140 | 0.020 | 0.060 | 0.060 | 0.340 |
| Doubao-1-5-thinking-pro | 0.160 | 0.600 | 0.860 | 0.800 | 0.880 | 0.640 | 0.800 | 0.920 | 0.340 | 0.980 | 0.680 |
| DeepSeek-R1-Distill-Qwen-32B | 0.120 | 0.200 | 0.260 | 0.220 | 0.400 | 0.540 | 0.120 | 0.100 | 0.020 | 0.700 | 0.380 |
| DeepSeek-R1-Distill-Qwen-7B | 0.060 | 0.000 | 0.060 | 0.000 | 0.120 | 0.020 | 0.020 | 0.000 | 0.000 | 0.140 | 0.000 |
| DeepSeek-R1 | 0.200 | 0.420 | 0.540 | 0.680 | 0.740 | 0.520 | 0.620 | 0.700 | 0.380 | 0.940 | 0.460 |
| DeepSeek-v3-0324 | 0.120 | 0.020 | 0.300 | 0.460 | 0.260 | 0.140 | 0.200 | 0.160 | 0.200 | 0.140 | 0.320 |
| Claude-3.7 | 0.040 | 0.140 | 0.060 | 0.060 | 0.340 | 0.280 | 0.180 | 0.120 | 0.180 | 0.200 | 0.700 |
| Claude-3.7-thinking | 0.080 | 0.100 | 0.160 | 0.640 | 0.720 | 0.880 | 0.480 | 0.500 | 0.680 | 0.700 | 0.000 |
| GPT-4o | 0.000 | 0.060 | 0.040 | 0.000 | 0.140 | 0.080 | 0.020 | 0.020 | 0.020 | 0.020 | 0.280 |
| O3-mini | 0.160 | 0.420 | 0.980 | 0.720 | 0.820 | 0.780 | 0.600 | 0.860 | 0.760 | 0.960 | 0.280 |
| O1-2024-12-17 | 0.260 | 0.040 | 1.000 | 0.700 | 0.840 | 0.600 | 0.340 | 0.280 | 0.760 | 0.800 | 0.300 |
| Gemini-2.0-Flash | 0.020 | 0.160 | 0.040 | 0.060 | 0.160 | 0.280 | 0.200 | 0.000 | 0.000 | 0.080 | 0.440 |
| Gemini-2.5-pro-03-25 | 0.240 | 0.640 | 0.800 | 0.760 | 0.780 | 0.920 | 0.540 | 0.920 | 1.000 | 0.980 | 0.840 |
| Gemini-2.0-Flash-thinking | 0.140 | 0.100 | 0.120 | 0.140 | 0.220 | 0.560 | 0.140 | 0.040 | 0.320 | 0.740 | 0.500 |
| Qwen-QwQ | 0.080 | 0.080 | 0.080 | 0.300 | 0.240 | 0.020 | 0.020 | 0.000 | 0.000 | 0.340 | 0.000 |
| Qwen2.5-72B-Instruct | 0.000 | 0.020 | 0.060 | 0.000 | 0.120 | 0.000 | 0.060 | 0.060 | 0.020 | 0.040 | 0.000 |
| Qwen2.5-32B-Instruct | 0.000 | 0.040 | 0.060 | 0.000 | 0.120 | 0.060 | 0.060 | 0.000 | 0.000 | 0.020 | 0.000 |
| Qwen2.5-7B-Instruct | 0.000 | 0.000 | 0.000 | 0.000 | 0.000 | 0.000 | 0.000 | 0.000 | 0.000 | 0.000 | 0.020 |

Table 6: Control, interaction, and task reasoning abilities of different models on KORGym.

| Model | Minigrid | Snake | Tetris | Tower of Hanoi | Numeric Bricks | Minesweeper | Nullify | PVZ | Long Cat | Black White Copy |
|---|---|---|---|---|---|---|---|---|---|---|
| Doubao-1.5-pro | 0.050 | 1.800 | 0.000 | 0.150 | 0.000 | 0.000 | 0.250 | 17.550 | 0.020 | 0.000 |
| Doubao-1-5-thinking-pro | 0.300 | 16.300 | 0.300 | 0.650 | 0.000 | 0.076 | 0.400 | 59.950 | 0.600 | 0.360 |
| DeepSeek-R1-Distill-Qwen-32B | 0.100 | 7.480 | 0.300 | 0.700 | 0.000 | 0.162 | 0.300 | 31.750 | 0.220 | 0.200 |
| DeepSeek-R1-Distill-Qwen-7B | 0.120 | 0.150 | 0.000 | 0.050 | 0.000 | 0.042 | 0.100 | 11.950 | 0.000 | 0.000 |
| DeepSeek-R1 | 0.100 | 16.500 | 0.650 | 0.700 | 0.020 | 0.185 | 0.200 | 68.450 | 0.480 | 0.300 |
| DeepSeek-v3-0324 | 0.250 | 4.250 | 0.050 | 0.200 | 0.000 | 0.063 | 0.300 | 20.350 | 0.300 | 0.080 |
| Claude-3.7 | 0.150 | 6.900 | 0.100 | 0.100 | 0.000 | 0.049 | 0.150 | 27.300 | 0.140 | 0.020 |
| Claude-3.7-thinking | 0.350 | 9.750 | 0.150 | 0.500 | 0.000 | 0.297 | 0.250 | 40.300 | 0.320 | 0.180 |
| GPT-4o | 0.000 | 0.600 | 0.000 | 0.050 | 0.000 | 0.027 | 0.150 | 25.000 | 0.000 | 0.000 |
| O3-mini | 0.300 | 15.950 | 0.900 | 1.000 | 0.000 | 0.555 | 0.250 | 41.200 | 0.700 | 0.360 |
| O1-2024-12-17 | 0.150 | 17.500 | 0.550 | 0.850 | 0.000 | 0.741 | 0.300 | 54.850 | 0.840 | 0.360 |
| Gemini-2.0-Flash | 0.150 | 2.050 | 0.000 | 0.100 | 0.000 | 0.008 | 0.300 | 19.650 | 0.100 | 0.020 |
| Gemini-2.5-pro-03-25 | 0.400 | 13.500 | 0.050 | 0.650 | 0.000 | 0.235 | 0.300 | 61.300 | 0.600 | 0.380 |
| Gemini-2.0-Flash-thinking | 0.100 | 1.850 | 0.050 | 0.200 | 0.000 | 0.002 | 0.250 | 26.000 | 0.080 | 0.100 |
| Qwen-QwQ | 0.150 | 3.950 | 0.000 | 0.600 | 0.000 | 0.052 | 0.350 | 30.750 | 0.020 | 0.160 |
| Qwen2.5-72B-Instruct | 0.050 | 0.620 | 0.000 | 0.200 | 0.000 | 0.000 | 0.250 | 24.850 | 0.040 | 0.000 |
| Qwen2.5-32B-Instruct | 0.050 | 0.300 | 0.000 | 0.050 | 0.000 | 0.021 | 0.200 | 14.450 | 0.020 | 0.020 |
| Qwen2.5-7B-Instruct | 0.000 | 0.450 | 0.000 | 0.050 | 0.000 | 0.000 | 0.250 | 0.050 | 0.000 | 0.020 |

Table 7: Language and textual reasoning abilities of different models on KORGym.

| Model | Word Problem | Alphabetical sorting | Letter Connection | Word Transformation | Wordle | Crypto Word |
|---|---|---|---|---|---|---|
| Doubao-1.5-pro | 0.120 | 0.360 | 0.160 | 0.140 | 0.100 | 0.150 |
| Doubao-1-5-thinking-pro | 0.600 | 0.880 | 0.720 | 0.400 | 0.600 | 1.000 |
| DeepSeek-R1-Distill-Qwen-32B | 0.340 | 0.500 | 0.220 | 0.140 | 0.450 | 0.000 |
| DeepSeek-R1-Distill-Qwen-7B | 0.020 | 0.240 | 0.020 | 0.000 | 0.050 | 0.000 |
| DeepSeek-R1 | 0.820 | 0.960 | 0.900 | 0.420 | 0.600 | 0.950 |
| DeepSeek-v3-0324 | 0.460 | 0.840 | 0.420 | 0.380 | 0.500 | 0.500 |
| Claude-3.7 | 0.580 | 0.840 | 0.660 | 0.220 | 0.250 | 0.650 |
| Claude-3.7-thinking | 0.820 | 0.980 | 0.960 | 0.560 | 0.850 | 1.000 |
| GPT-4o | 0.420 | 0.340 | 0.160 | 0.120 | 0.400 | 0.100 |
| O3-mini | 0.880 | 0.980 | 0.980 | 0.400 | 0.400 | 1.000 |
| O1-2024-12-17 | 0.960 | 1.000 | 0.980 | 0.480 | 0.450 | 0.850 |
| Gemini-2.0-Flash | 0.340 | 0.560 | 0.340 | 0.120 | 0.250 | 0.100 |
| Gemini-2.5-pro-03-25 | 0.900 | 0.960 | 0.940 | 0.780 | 0.650 | 1.000 |
| Gemini-2.0-Flash-thinking | 0.620 | 0.780 | 0.400 | 0.180 | 0.550 | 0.500 |
| Qwen-QwQ | 0.480 | 0.760 | 0.400 | 0.180 | 0.400 | 0.050 |
| Qwen2.5-72B-Instruct | 0.080 | 0.280 | 0.160 | 0.020 | 0.200 | 0.000 |
| Qwen2.5-32B-Instruct | 0.100 | 0.280 | 0.080 | 0.000 | 0.050 | 0.050 |
| Qwen2.5-7B-Instruct | 0.040 | 0.140 | 0.020 | 0.000 | 0.050 | 0.000 |

Table 8: Spatial and geometric reasoning abilities of different models on KORGym.

| Model | Maze | Sokoban | play lines | Emoji Connect | 8-puzzle | Bubble Ball Sorting | Pipe Game | Free the Key | Map Simulation | Arrow Pathway |
|---|---|---|---|---|---|---|---|---|---|---|
| Doubao-1.5-pro | 0.000 | 0.000 | 0.080 | 0.080 | 0.080 | 0.300 | 0.000 | 0.050 | 0.020 | 0.000 |
| Doubao-1-5-thinking-pro | 0.800 | 0.420 | 0.580 | 0.580 | 0.680 | 1.000 | 0.700 | 0.950 | 0.320 | 0.000 |
| DeepSeek-R1-Distill-Qwen-32B | 0.380 | 0.160 | 0.140 | 0.220 | 0.380 | 0.900 | 0.040 | 0.700 | 0.040 | 0.000 |
| DeepSeek-R1-Distill-Qwen-7B | 0.120 | 0.040 | 0.000 | 0.000 | 0.100 | 0.000 | 0.000 | 0.000 | 0.000 | 0.000 |
| DeepSeek-R1 | 0.600 | 0.440 | 0.360 | 0.760 | 0.520 | 1.000 | 0.300 | 0.750 | 0.140 | 0.000 |
| DeepSeek-v3-0324 | 0.380 | 0.080 | 0.080 | 0.560 | 0.220 | 0.250 | 0.020 | 0.650 | 0.060 | 0.000 |
| Claude-3.7 | 0.020 | 0.020 | 0.020 | 0.160 | 0.140 | 0.850 | 0.000 | 0.200 | 0.040 | 0.020 |
| Claude-3.7-thinking | 0.400 | 0.260 | 0.240 | 0.760 | 0.460 | 0.950 | 0.100 | 0.850 | 0.120 | 0.180 |
| GPT-4o | 0.020 | 0.040 | 0.020 | 0.120 | 0.040 | 0.450 | 0.000 | 0.300 | 0.000 | 0.000 |
| O3-mini | 0.860 | 0.780 | 0.720 | 0.960 | 0.940 | 0.950 | 0.540 | 0.850 | 0.380 | 0.300 |
| O1-2024-12-17 | 0.360 | 0.700 | 0.840 | 0.940 | 0.900 | 0.950 | 0.840 | 0.850 | 0.280 | 0.000 |
| Gemini-2.0-Flash | 0.020 | 0.000 | 0.000 | 0.120 | 0.040 | 0.600 | 0.000 | 0.050 | 0.100 | 0.000 |
| Gemini-2.5-pro-03-25 | 0.500 | 0.100 | 0.660 | 0.620 | 0.720 | 0.950 | 0.740 | 0.850 | 0.100 | 0.000 |
| Gemini-2.0-Flash-thinking | 0.020 | 0.060 | 0.000 | 0.180 | 0.480 | 0.650 | 0.000 | 0.200 | 0.020 | 0.020 |
| Qwen-QwQ | 0.400 | 0.180 | 0.080 | 0.060 | 0.240 | 0.000 | 0.000 | 0.650 | 0.000 | 0.000 |
| Qwen2.5-72B-Instruct | 0.040 | 0.020 | 0.020 | 0.060 | 0.040 | 0.350 | 0.000 | 0.100 | 0.000 | 0.000 |
| Qwen2.5-32B-Instruct | 0.020 | 0.000 | 0.040 | 0.000 | 0.000 | 0.500 | 0.000 | 0.200 | 0.020 | 0.000 |
| Qwen2.5-7B-Instruct | 0.000 | 0.000 | 0.000 | 0.020 | 0.000 | 0.000 | 0.000 | 0.000 | 0.000 | 0.000 |

Table 9: Strategic and game-theoretic reasoning abilities of different models on KORGym.

| Model | 2048 | Trust Evolution | N point | Spider Solitaire | Circle the cat |
|---|---|---|---|---|---|
| Doubao-1.5-pro | 648.800 | 31.650 | 7.500 | 0.000 | 0.000 |
| Doubao-1-5-thinking-pro | 1736.000 | 36.600 | 8.000 | 0.000 | 0.050 |
| DeepSeek-R1-Distill-Qwen-32B | 990.000 | 37.700 | 8.050 | 0.000 | 0.050 |
| DeepSeek-R1-Distill-Qwen-7B | 884.200 | 36.350 | 3.625 | 0.000 | 0.000 |
| DeepSeek-R1 | 2221.800 | 63.500 | 8.250 | 0.000 | 0.200 |
| DeepSeek-v3-0324 | 1101.400 | 37.750 | 8.225 | 0.000 | 0.200 |
| Claude-3.7 | 809.000 | 36.900 | 8.300 | 0.000 | 0.000 |
| Claude-3.7-thinking | 1283.600 | 37.950 | 8.200 | 0.000 | 0.100 |
| GPT-4o | 958.600 | 54.750 | 7.650 | 0.000 | 0.000 |
| O3-mini | 1285.600 | 34.550 | 7.825 | 0.000 | 0.300 |
| O1-2024-12-17 | 1652.200 | 64.500 | 8.600 | 0.000 | 0.000 |
| Gemini-2.0-Flash | 879.800 | 51.750 | 7.475 | 0.000 | 0.000 |
| Gemini-2.5-pro-03-25 | 2083.200 | 66.500 | 7.750 | 0.000 | 0.250 |
| Gemini-2.0-Flash-thinking | 1378.200 | 37.500 | 7.925 | 0.000 | 0.000 |
| Qwen-QwQ | 681.800 | 58.400 | 3.950 | 0.000 | 0.000 |
| Qwen2.5-72B-Instruct | 773.400 | 47.150 | 7.600 | 0.000 | 0.000 |
| Qwen2.5-32B-Instruct | 660.000 | 51.850 | 7.550 | 0.000 | 0.000 |
| Qwen2.5-7B-Instruct | 780.600 | 0.000 | 6.200 | 0.000 | 0.050 |

# E  PCA Result

The first two principal components capture 96.2% of the total variance (PC1: 91.9%, PC2: 4.3%), as shown in Figure 5b. The model PCA cluster results are as follows:

- **Cluster 0**:Doubao-1-5-thinking-pro, DeepSeek-R1, Claude-3.7-thinking, Gemini-2.5-pro-03-25, Qwen3-32B

- **Cluster 1**:O3-mini, O1-2024-12-17

- **Cluster 2**:Doubao-1.5-pro, DeepSeek-R1-Distill-Qwen-7B, GPT-4o, Gemini-2.0-Flash, Qwen2.5-72B-Instruct, Qwen2.5-32B-Instruct, Qwen2.5-7B-Instruct

- **Cluster 3**:: DeepSeek-R1-Distill-Qwen-32B, DeepSeek-v3-0324, Claude-3.7, Gemini-2.0-Flash-thinking, Qwen-QwQ

These results reveal the following insights from the clustering structure:

- **Cluster 1 predominantly consists of the GPT-o series reasoning models**, which exhibit highly similar reasoning patterns, indicating a consistent architectural behavior within the series.

- **Cluster 0 includes top-performing reasoning models**, such as Claude-3.7, Gemini-2.5, and DeepSeek-R1, which show strong and mutually consistent performance across dimensions.

- **Qwen3 and Doubao-1.5-thinking-pro are located near the boundary between Cluster 0 and Cluster 1**, suggesting that these models share reasoning characteristics with both the GPT-o series and the leading reasoning models in Cluster 0.
- **Cluster 3 primarily contains a mix of open-source reasoning models and closed-source non-reasoning models**, exhibiting moderate overall reasoning ability.
- **Cluster 2 consists of mainstream open-source non-reasoning models and closed-source baseline APIs**, representing a group of lower-performing or general-purpose models.

# F Detailed Result of Impact of Disabling Each Paradigm on Model Performance

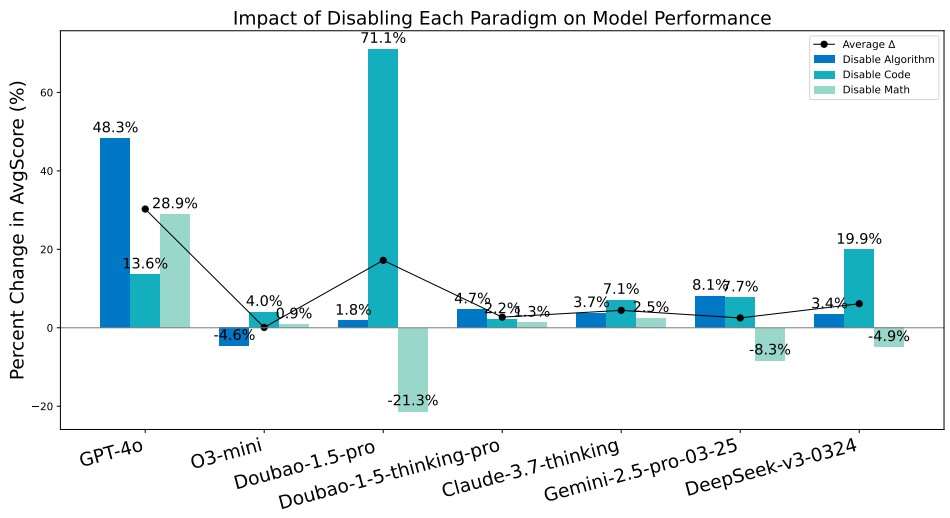

Figure 8: Detailed Result of Impact of Disabling Each Paradigm on Model Performance.

# G Comparison of different game benchmarks.

Table 10: Comparison of different game benchmarks.

| Benchmark | # Games | Multi-turn | RL | Controllable Difficulty | Multimodal |
|---|---|---|---|---|---|
| LogicGame [9] | 27 | ✗ | ✗ | ✗ | ✗ |
| AgentBench [26] | 8 | ✓ | ✗ | ✗ | ✗ |
| GameArena [27] | 3 | ✓ | ✗ | ✗ | ✗ |
| SPIN-Bench [28] | 6 | ✓ | ✓ | ✗ | ✗ |
| TEXTARENA [10] | 57 | ✓ | ✓ | ✗ | ✗ |
| ReasoningGYM [29] | 100+ | ✗ | ✓ | ✓ | ✗ |
| KORGym | 51 | ✓ | ✓ | ✓ | ✓ |

# H   Detailed Game Category

Table 11: Game Introduction of KORGym. From top to bottom, the categories are: Mathematical and Logical Reasoning games, Puzzle Reasoning games, Spatial and Geometric Reasoning games, Strategic Reasoning games, Control and Interaction Reasoning games, and Multimodal Reasoning games.

| Name | Task Content |
|---|---|
| Crossword Puzzle | This game challenges LLM to infer current date given a future date and number of days between them. |
| Sudoku | This game evaluates an LLM's logical reasoning by solving a Sudoku puzzle. |
| Light Out Game | This game tests an LLM's strategic reasoning by requiring it to switch off all lights on a 3x3 grid. |
| Square Addition | This game requires LLM to compute column sums based on symbolic values. |
| Alien | This game requires LLM to count alien species based on multiple traits from a complex dataset. |
| Party Time | This game challenges LLM to identify and count students who meet specific criteria. |
| Path Planning Problem | This game requires LLM to calculate the shortest distance between two cities within a complex network. |
| Construction Company | This game challenges LLM to calculate the minimum time across multiple companies and projects. |
| Tower of Hanoi | This game requires LLM to solve a Tower of Hanoi puzzle, moving disks between columns to reach a goal state. |
| Numeric Bricks | This game challenges an LLM to fill a grid by expanding each labeled cell according to a specified count. |
| One Stroke Drawing | This game requires LLM to find an Eulerian path that visits every edge exactly once. |
| Nullify | This game challenges an LLM to combine arithmetic units using operations to achieve a final result of zero. |
| Coloring Issue | This game challenges an LLM to assign colors to graph nodes such that no two adjacent nodes share the same color. |
| City Traveller | This game requires LLM to extract, filter, and compute city information from a complex city network. |
| Word Problem | This game challenges an LLM to find a specific English word that matches several constraints. |
| Alphabetical Sorting | This game requires LLM to rearrange the remaining unordered letters to form a valid word. |
| Letter Connection | This game requires LLM to reconstruct a hidden word from a 3x3 letter grid. |
| Word Transformation | This game requires LLM to decode a transformed word by reasoning through layered transformations . |
| Wordle | This game requires LLM to perform deductive reasoning through iterative word guessing based on structured feedback. |
| Crypto Word | This game requires LLM to decode a sentence by mapping emojis to letters through iterative feedback-based guessing. |
| Maze | This game requires LLM to find a valid path through a maze from the start point to the destination. |
| Sokoban | This game requires LLM to solve a Sokoban puzzle, pushing all boxes onto the target areas. |
| Playlines | This game requires LLM to fill in grids by connecting identical numbers on a grid without leaving empty spaces . |
| Emoji Connect | This game requires LLM to count repeated horizontal or vertical patterns in emoji grids. |
| 8-puzzle | This game requires LLM to plan valid tile moves to reposition a target tile in a sliding puzzle grid. |
| Bubble Ball Sorting | This game requires LLM to sort colored balls into uniform tubes under stacking constraints. |
| Pipe Game | This game requires LLM capability to rotate pipes in a grid to create a continuous path. |
| Free the Key | This game requires LLM to move the building blocks and keys to reach the exit. |
| Map Simulation | This game tests an LLM's ability to simulate multi-step movement through a dynamic grid. |
| Arrow-pathway | This game tests an LLM's ability to navigate a maze by sequencing directional actions to trigger waypoints. |
| 2048 | This game requires LLM to play the 2048 puzzle by choosing the best move based on the current game board |
| Trust Evolution | This game requires LLM to identify and exploit opponent behavior patterns through strategic decision-making . |
| N-point | This game requires LLM to play expanded 21-Point in dynamic thresholds and an opponent's fixed behavior. |
| Spider Solitaire | This game tests an LLM's ability to plan and execute strategic moves in Spider Solitaire. |
| Circle the Cat | This game requires LLM to strategically place walls on a hexagonal grid to trap a moving cat. |
| Minigrid | This game requires LLM to solve a series of tasks based on the Minigrid [30] reinforcement learning system. |
| Snake | This game requires LLM to control a growing snake on a bounded grid, avoiding collisions to maximize score. |
| Tetris | This requires LLM to plan by strategically rotating and placing Tetris blocks to clear rows and maximize score. |
| Minesweeper | This game requires LLM to uncover safe cells and flag hidden mines . |
| PVZ | This game requires LLM to place plants to counter increasingly strong zombies. |
| Long Cat | This game requires LLM to plan efficient movement sequences by navigating a sliding cat to fill all empty spaces. |
| Black White Copy | This game requires LLM to toggle rows to transform the board into a specified black-and-white target pattern. |
| Crossword Puzzle | This game requires LLM to solve linguistic clues to fill the grid with words correctly. |
| Jigsaw Puzzle | This game requires LLM to match visual puzzle pieces with numbered slots . |
| Find The Pattern | This game requires LLM to identify the correct visual piece that completes a given pattern. |
| Circle The Cat (Visual) | This game requires LLM to analyze a visual board and determine optimal wall placements to prevent a cat. |
| Map Simulation (Visual) | This game requires LLM to interact with diverse objects, and accurately calculate the final position. |
| Sokoban (Visual) | This game requires LLM to interpret a visual Sokoban puzzle and generate a precise series of moves. |
| Bubble Ball Sorting (Visual) | This game requires LLM to generate valid moves to achieve uniform color sorting across specified tubes. |
| Wordle (Visual) | This game requires LLM to deduce the correct secret word through multiple turns of guessing. |
| Square Addition (Visual) | This game requires LLM to infer integer values to compute accurate column sums. |

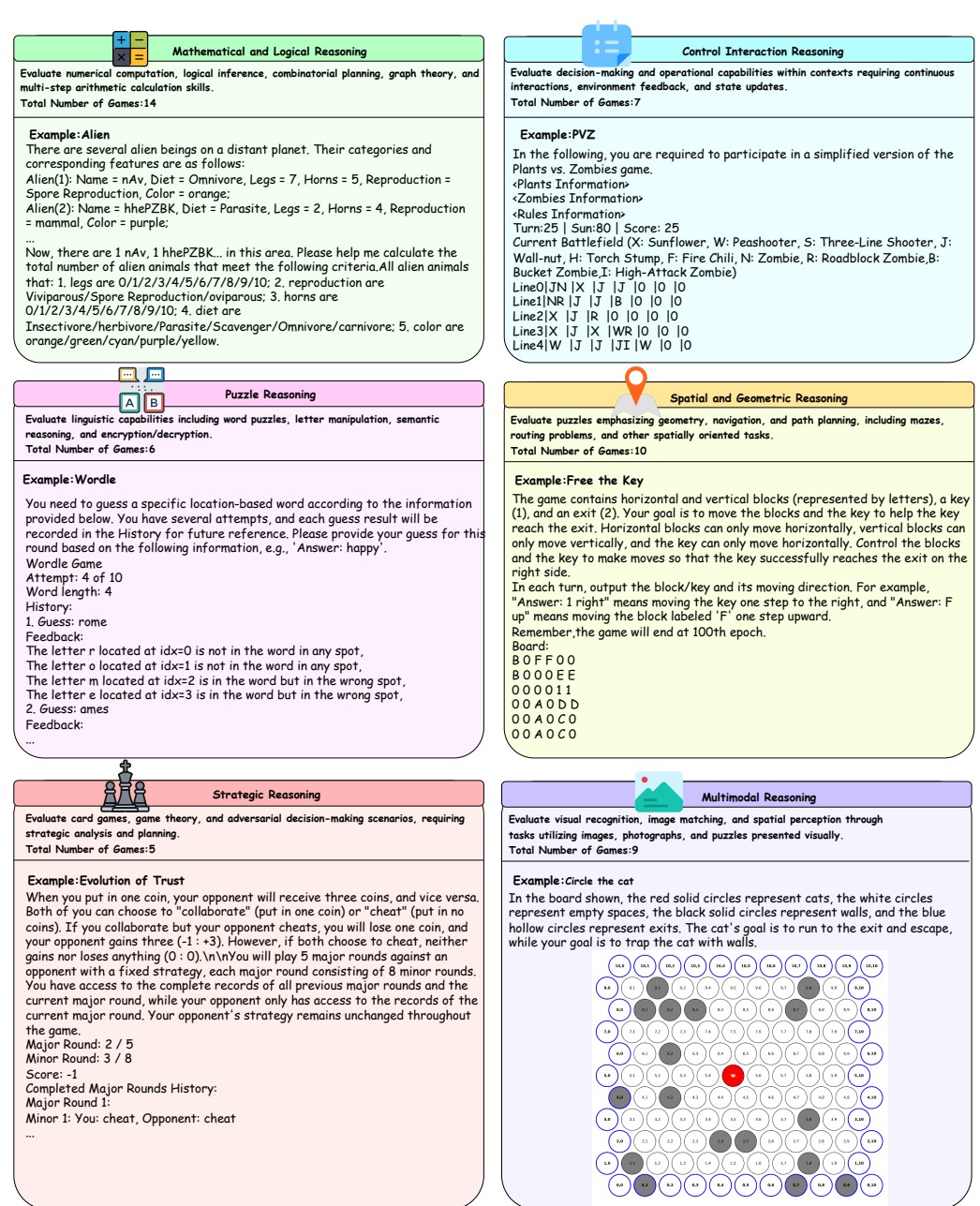

Figure 9: Overview of the KORGym tasks. Our KORGym supports over 50 novel games, enabling precise and efficient evaluation of large language models (LLMs) across six distinct capability dimensions.

# I   Models Evaluated in KORGym

Table 12: Models evaluated in KORGym.

| Series | Model |
|---|---|
| GPT | GPT-4o [31], O1 [32], O3-mini[33] |
| Claude | Claude-3.7-Sonnet [34],Claude-3.7-Sonnet-thinking |
| Doubao | Doubao-1.5-pro [35], Doubao-1.5-thinking-pro [36], Doubao-vision-pro [37] |
| DeepSeek | DeepSeek-v3 [38], DeepSeek-R1 [39], DeepSeek-R1-Distill-Qwen(7B,32B) |
| Qwen | Qwen3-32B [40], Qwen2.5-Instruct(7B,32B,72B) [41], Qwen-QwQ-32B [42], Qwen2.5-VL-Instruct(7B,32B,72B) [43] |
| Gemini | Gemini-2.0-Flash [44], Gemini-2.0-Flash-thinking,Gemini-2.5-pro [45] |
| InternVL | InternVL3-68B [46] |

# J   Detailed Result of Textual and Multimodal Versions of the Same Game

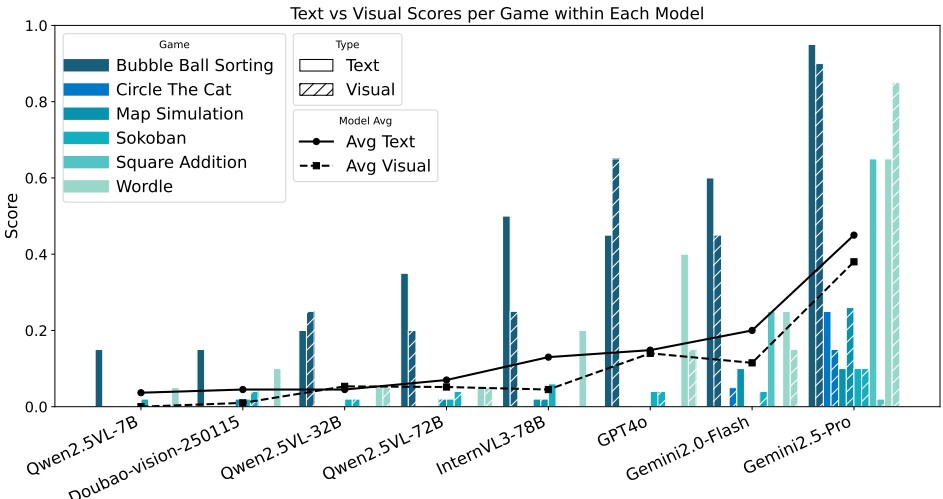

Figure 10: Performance Comparison of the Same Model on Textual and Multimodal Versions of the Same Game. In this visualization, bar colors indicate different games, while shading of the bars distinguishes between text (no shading) and visual (with shading) versions. Solid and dashed lines represent the average scores for the textual and visual versions, respectively.

