# OpenReview forum: "KORGym: A Dynamic Game Platform for LLM Reasoning Evaluation"
_NeurIPS.cc/2025/Conference — NeurIPS 2025 spotlight_

### Official Review · Reviewer_SShB · 2025-07-02

**Clarity:** 3
**Significance:** 3
**Originality:** 3
**Rating:** 5
**Confidence:** 3

**Summary:**

This paper present KORGym, a new benchmark of over fifty games targeting knowledge-orthogonal evaluation of LLM reasoning abilities. The benchmark categorizes the games along six reasoning dimensions, and supports an aggregated mean metric providing a normalized capability-wise evaluation of reasoning. The paper provides evaluation of many widely used LLMs, and informative analyses are provided.

**Questions:**

- Could the authors show correlation with other tasks or benchmarks which feature more difficult or specialized tasks? For example control and interaction could measure correlation with alfworld or minecraft.
- How is the reasoning paradigm disabling achieved?

**Ethical Concerns:**

["NO or VERY MINOR ethics concerns only"]

**Final Justification:**

The rebuttal addressed my concerns, which were about 1) the core claim of knowledge-orthogonality of the games, and 2) the lack of popular complex benchmarks for agents like Minecraft. The author response shows that the benchmark is overall sufficiently knowledge-orthogonal and leakage happens in certain recent models like o3-family, which is interesting in itself, and likely to provide new insights into the performance of latest agents. The authors also show a correlation study indicating that, taken together the benchmark could capture capabilities required for more complex benchmarks like Minecraft. Overall the paper makes meaningful contributions in improving the evaluation of reasoning agents, so I have give an accept rating.

**Limitations:**

yes

**Quality:**

3

**Strengths And Weaknesses:**

Strengths
- The benchmark is well motivated and well designed. Due to many existing benchmarks becoming saturated due to pretrained knowledge leakage, it has value in helping assess the raw reasoning capability of models.
- The benchmark is not saturated, particularly by open-source models, and it could also be interesting to see agent scaffolds evaluated on this benchmark.
- The analyses of reasoning paradigms is particularly interesting, showing that model series' exhibit paradigm preferences, and some reasoning paradigms like code-based reasoning, can even partially constrain performance. This also enables interesting observations e.g. that the strongest model tested, o3-mini, does not exhibit this bias.

Weaknesses
- The core claim that the benchmark is knowledge-orthogonal is not validated.
- The control, interaction, and task reasoning part of the benchmark feels weak, as it consists of relatively simple tasks in terms of control and interaction complexity.

---

> ### Author Rebuttal · Authors · 2025-07-28
>
> Thank you very much for your valuable feedback and suggestions on KORGYM. We sincerely apologize for the shortcomings in our work and have conducted a series of experiments and improvements to address these concerns:
>
> ---
>
> **Response to Weakness 1**
>
> (1) **Question 1**:The core claim that the benchmark is knowledge-orthogonal is not validated.
>
> (1) **Answer 1**:
>
> Although we formally defined the concept of KOR in Appendix C of our submitted paper, it did not permit empirical quantification of pre-trained knowledge leakage, thus failing to validate our core claim of knowledge orthogonality. To rectify this, we have extended the formal definition and introduced the Knowledge Leakage Coefficient, which can be empirically calculated through experiments, to further substantiate the validity of KORGYM.
>
> **1.1 Notational Definitions**
>
> For a game within KORGYM, the required reasoning information comprises:
>
> - **$K$**: General background or domain‑specific knowledge acquired during pre‑training and post-training.
> - **$T$**: Single‑turn or multi‑turn tasks that must be completed during gameplay.
> - **$R$**: Core game‑rule information specifically designed for solving task $T$.
> - **$A$**: The set of actions generated by the LLM to solve task $T$.
> - **$\rightarrow$**: The reasoning and interaction process whereby the LLM gradually derives the action set $A$ from the game task $T$.
> - **$S$**: The final score obtained by the LLM's interaction with the game environment based on reasoning from $R$ and/or $K$:
>   - $S(T \rightarrow A \mid K)$: Score from actions derived solely from pre‑trained knowledge $K$.
>   - $S(T \rightarrow A \mid R, K)$: Score from actions derived using both game‑rule knowledge $R$ and pre‑trained knowledge $K$.
>
> ---
>
> **1.2 Analysis of Game Redundancy**
>
> 1. **Knowledge‑Rule Decoupling**
>    Ideally, the game rule information $R$ is logically self‑contained and independent of pre‑trained knowledge $K$:
>    $$
>    R \perp K
>    $$
>
> 2. **Knowledge Leakage Coefficient**
>    The Leakage between $K$ and $R$ is quantified by the Knowledge Impact Coefficient $\beta$:
>    $$
>    \beta
>    = \frac{S(T \rightarrow A \mid K)}{S(T \rightarrow A \mid R, K)},
>    \quad \beta \in [0,1].
>    $$
>   A smaller $\beta$ indicates less overlap between the LLM’s pre-trained knowledge and the game rules. Consequently, the LLM cannot achieve optimal scores without explicitly relying on the game rules provided in the prompt.
>
>
> ---
>
> **1.3 Quantitative Experiment on Knowledge Leakage**
>
> Based on the above definition, we select a subset of games from KORGYM, including some classic games (Snake, Sokoban, and 8-puzzle) and original/adapted games (Pipe Game, Long Cat, ...). We conduct an ablation study in which the core game‑rule information (R) was removed and then compute the $\beta$:
>
>
> |    Model    |  Snake  | Pipe Game | Long Cat | Sokoban | Word Transformation | 8‑puzzle | Play Lines | Black White Copy |
> |:-----------:|:-------:|:---------:|:--------:|:--------:|:-------------------:|:--------:|:----------:|:----------------:|
> | Doubao-1-5-thinking-pro |  0.055  |   0.085   |    0     |  0.904  |         0           |  0.176   |     0      |      0.166       |
> | DeepSeek-R1             |  0.333  |   0.200   |   0.208  |  0.636  |         0           |  0.307   |     0      |      0.133       |
> | Qwen3-32B-thinking      |  0.071  |   0.080   |    0     |  0.909  |         0           |  0.321   |     0      |      0.117       |
> | o3‑mini                 |  0.956  |   0.000   |    0     |  0.948  |         0           |  0.617   |     0      |      0.166       |
> | Claude-3.7-thinking     |  0.441  |   0.200   |   0.312  |  0.461  |         0           |  0.521   |     0      |      0.222       |
> | Average | 0.371 | 0.113 | 0.104 | 0.771 | 0 | 0.388 | 0 | 0.160 |
>
>
> >From the above results, we can draw the following preliminary conclusions:
>
> > - **Model perspective:** Models such as **o3‑mini** and **DeepSeek‑R1** exhibit relatively notable knowledge leakage, whereas other models (e.g., **Doubao‑1‑5‑thinking‑pro**) demonstrate comparatively lower leakage. This observation supports the view that **RL processes genuinely enhance a model's reasoning capabilities, rather than merely enabling it to memorize training questions**.
> > -  **Game perspective:** Knowledge leakage is more prominent in **classic games**, while the corresponding β values for **original/adapted games** remain at a relatively low level. Given that more than **80%** of the games in KORGYM are originally developed or adapted by our engineering team, this strongly supports **the validity and reliability of KORGYM**.
> > -  **Methodological perspective:** We believe that the knowledge leakage quantification experiment itself offers an interesting insight and methodology for **analyzing models’ genuine reasoning abilities and uncovering information contained within the pre-training data**.
>
> Thus, we have demonstrated the effectiveness of KORGYM in achieving knowledge orthogonality. Additionally, we plan to introduce more original and adapted games in the future and exclude games with higher β values to further enhance the degree of knowledge orthogonality in KORGYM.
>
> ---
>
> **Response to Weakness 2**
>
> (2) **Question 2**:The control, interaction, and task reasoning part of the benchmark feels weak, as it consists of relatively simple tasks in terms of control and interaction complexity.
>
> (2) **Answer 2**:
>
> Due to time constraints and considerations for overall system coherence, we regretfully did not include highly complex games, such as Minecraft, within the control, interaction, and task reasoning components.  We aimed to address these reasoning dimensions by decomposing them into specific sub-capabilities, each assessed through dedicated games in KORGYM. For example:
>
> > -  **Minigrid** evaluates the model's capability to interact with environments using tools.
> > -  **PVZ** assesses the model's capacity for resource management and allocation (e.g., sunlight) within dynamic and continuous environments, as well as its multi-step strategic planning based on opponent progression.
> > -  **Minesweeper** examines logical reasoning, probabilistic estimation, and risk assessment abilities under incomplete information.
>
> By decomposing these dimensions into sub-capabilities and designing corresponding games, we provided a comprehensive evaluation framework for control, interaction, and task reasoning. Furthermore, in *Response to Question 1*, we quantitatively demonstrated the effectiveness of this approach through Spearman coefficient analysis.
>
> ---
>
> **Response to Question 1**
>
> (3) **Question 3**: Could the authors show correlation with other tasks or benchmarks which feature more difficult or specialized tasks? For example control and interaction could measure correlation with alfworld or minecraft.
>
> (3) **Answer 3**:
>
> To further analyze the correlation with other tasks or benchmarks featuring more difficult or specialized tasks, we conducted a Spearman correlation analysis with the following domain-specific benchmarks/projects:
>
> > -  MathArena: Specialized assessment of model capabilities in mathematics;
> > -  BBEH: An extended benchmark for BBH proposed by Google;
> > -  MC Bench: Evaluates the model's ability to construct structures described in prompts using Minecraft blocks;
> > -  OmniSpatial: A benchmark designed for comprehensive spatial reasoning evaluation;
> > -  lmARENA-Vision: A comprehensive benchmark assessing visual reasoning capabilities;
> > -  Ace Attorney : A project leveraging the classic game Ace Attorney to test interaction and reasoning capabilities of LLMs;
> > -  BizFinBench: A benchmark for evaluating long-term strategic planning capabilities in financial contexts;
> > -  SnakeBench: A project employing LLM versus LLM scenarios to evaluate competitive gaming capabilities.
>
> The corresponding meanings of the Capability Dimension fields are as follows:
> > -  **MLR** : Mathematical and Logical Reasoning
> > -  **CIR** : Control Interaction Reasoning
> > -  **PR**: Puzzle Reasoning
> > -  **SGR**: Spatial and Geometric Reasoning
> > -  **SR**: Strategic Reasoning
> > -  **MR**: Multimodal Reasoning
>
> The results of our analysis are as follows:
>
>
> | Capability Dimension| MLR  |         MLR          | CIR |       CIR        |     PR       |   PR      | SGR |   SGR     | SR     | SR     | MR |
> |-------------|:---------------------------------:|:-------------------------------:|:----------------------------:|:-------------:|:---------------------------:|:-------:|:-----------------------------:|:----------:|:----------:|:----------:|:----------:|
> |   **Related Benchmark**    | MathArena                         | BBEH‑TimeArithmetic            | MC Bench                     | Ace Attorney  | BBEH‑Boolean Expressions    | BBEH‑BoardGameQA | BBEH‑Spatial Reasoning       | OmniSpatial | BizFinBench | SnakeBench | lmARENA-Vision |
> | **Spearman**| 0.866                             | 0.899                           | 0.828                        | 0.666         | 0.700                       | 0.600   | 0.899                         | 0.700      | 0.733 | 0.761 | 0.899|
>
> Thus, it can be inferred that KORGYM truly has a correlation with specialized tasks, and can reliably reflect the true reasoning capabilities of models across various dimensions.
>
> ---
>
> **Response to Question 2**
>
> (4) **Question 4**: How is the reasoning paradigm disabling achieved?
>
> (4) **Answer 4**:
>
> To implement the reasoning paradigm disabling, we added explicit instructions in the game rules prohibiting the use of specific reasoning paradigms as auxiliary methods. We also provided illustrative examples, such as:
> ```python
> "Remember, you cannot use the Code analysis method to help you solve the problem, e.g., "import math; a = 0; for i in range(...): ..." is forbidden."
> ```
> Finally, we sincerely thank you once again for your valuable feedback and suggestions on KORGYM, and we look forward to further discussions with you.

---

> > ### Comment · Reviewer_SShB · 2025-08-05
> >
> > I appreciate the authors' detailed response. The response address my concerns, and I have increased my rating accordingly.

---

> > > ### Author Response · Authors · 2025-08-06
> > > **Thank you for your insightful comments!**
> > >
> > > We are truly grateful for your thoughtful feedback and ongoing support for the acceptance of our paper. Your recognition and encouraging remarks are deeply valued, and they provide significant motivation for us to continue our research.

---

### Official Review · Reviewer_3dhT · 2025-07-03

**Clarity:** 4
**Significance:** 4
**Originality:** 4
**Rating:** 6
**Confidence:** 5

**Summary:**

This paper introduces a new benchmark, namely KORGym, which evaluates multiple dimensions of reasoning abilities in an LLM. KORGym is based on a dataset of over fifty games. KORGym consists of separate modules, mainly to parse inputs, initialise game environments and model inference. KORGym uses a detailed scoring scheme for different types of games, such as for single objective games, 1 point is assigned for success and 0 for failure, and for games that award incremental points, they accumulate all points. After this, KORGym uses "Capability Dimension Aggregated Mean" which is an aggregation metric that takes into account different ranges of scores and aggregates it semantically. KORGym provides results on many LLMs or all the reasoning tasks. Furthermore, provides some results on multimodal reasoning.

**Questions:**

None. I have no questions, as the paper is quite nicely presented. It would be nice to simplify the Figure 2. I understand the effort put into it. I also understand that there are a lot of pieces that are important and should be there. It would also be nice for the paper to include the above mentioned citations, plus more citations especially in LLMs for Gaming subsection, as the field has matured to a certain extent and many readers want to explore more papers through your paper. I am willing to change the score to strong accept as I foresee many readers coming to your paper, a lot of them will be skimming and they do need a simplified flow diagram and more papers to read from your paper.

**Ethical Concerns:**

["NO or VERY MINOR ethics concerns only"]

**Final Justification:**

I think the paper is a very solid improvement to the field. Hence, I have improved my scores.

**Limitations:**

Yes

**Quality:**

4

**Strengths And Weaknesses:**

**Strengths:**

- Very neatly written paper.
- Precise, to the point description of the framework.
- Brings in over 50 present games to test multiple reasoning fronts: mathematical and logical reasoning, control interaction reasoning, puzzle reasoning, spatial and geometric reasoning, strategic reasoning and multimodal reasoning.
- The evaluation metric is well-derived. Takes into account that different games have different scoring schemes. Provides fair evaluation scheme.

**Weaknesses:**

- Figure 2 is not straightforward to understand. I appreciate the effort put into it but simplifying it will help the paper.
- For related work I believe a few citations are a must: PlanBench and the works that followed it [1, 2], as it is one of the first works to evaluate reasoning in LLMs, and GameTraversalBenchmark [3], as it evaluates spatial reasoning and is the first benchmark to evaluate LLMs on games.

[1] Valmeekam, K., Marquez, M., Olmo, A., Sreedharan, S. and Kambhampati, S., 2023. Planbench: An extensible benchmark for evaluating large language models on planning and reasoning about change. Advances in Neural Information Processing Systems, 36, pp.38975-38987.

[2] Valmeekam, K., Stechly, K. and Kambhampati, S., 2024. LLMs Still Can't Plan; Can LRMs? A Preliminary Evaluation of OpenAI's o1 on PlanBench. arXiv preprint arXiv:2409.13373.

[3] Nasir, M.U., James, S. and Togelius, J., 2024. GameTraversalBenchmark: Evaluating Planning Abilities Of Large Language Models Through Traversing 2D Game Maps. arXiv preprint arXiv:2410.07765.

---

> ### Author Rebuttal · Authors · 2025-07-28
>
> We sincerely appreciate your recognition and support for our work, and we deeply apologize for the oversight regarding the Figure and related work sections in the paper.
>
> ---
>
> **Response to Weakness 1**
>
> (1) **Question 1**: Figure 2 is not straightforward to understand. I appreciate the effort put into it but simplifying it will help the paper.
>
> (1) **Answer 1**:
>
> We have simplified the content of the figure to enhance readability and provided additional explanations and supplements in the body text. We will update these adjustments once the paper is accepted.
>
> **Response to Weakness 2**
>
> ---
>
> (2) **Question 2**: For related work I believe a few citations are a must: PlanBench and the works that followed it [1, 2], as it is one of the first works to evaluate reasoning in LLMs, and GameTraversalBenchmark [3], as it evaluates spatial reasoning and is the first benchmark to evaluate LLMs on games.
>
> (2) **Answer 2**:
>
>
> We sincerely appreciate your insightful feedback on the shortcomings in our related work section and are grateful for recommending exceptionally engaging and inspiring studies. We found the PlanBench [1] and the accompanying Evaluation work [2], along with the GameTraversalBenchmark [3] that you suggested, to be truly fascinating and substantial contributions to the field. Reading these works significantly enriched our understanding and provided invaluable inspiration during the revision of our paper. We have eagerly integrated the relevant citations into our updated related work section, and we will reflect these enhancements comprehensively in the final version upon acceptance. Moreover, we intend to actively reference these insightful studies in our future research endeavors and surveys, thereby amplifying their visibility and sharing their valuable contributions with a broader scholarly audience.
>
> ---
>
> Once again, thank you very much for your support and constructive suggestions on KORGYM. We look forward to further discussions with you.

---

> > ### Comment · Reviewer_3dhT · 2025-08-05
> >
> > I sincerely thank the authors the authors for their improvements to their paper as per my suggestions. I think the paper is very solid improvement to the field. Hence, I have improved my scores.

---

> > > ### Author Response · Authors · 2025-08-06
> > > **Thank you for your insightful comments！**
> > >
> > > Thank you very much for your insightful comments and your continued support for our paper's acceptance. Your acknowledgment and encouragement are deeply appreciated, and your positive feedback significantly inspires us to advance our research efforts.

---

### Official Review · Reviewer_V7Eo · 2025-07-03

**Clarity:** 4
**Significance:** 4
**Originality:** 4
**Rating:** 6
**Confidence:** 5

**Summary:**

The paper introduces KORGym, a novel interactive benchmark designed to evaluate the reasoning capabilities of large language models (LLMs). KORGym consists of 50 games, most of which do not require external knowledge, hence the term Knowledge Orthogonal. The benchmark includes both textual and visual games. A wide range of LLMs is evaluated on this benchmark, and the results provide valuable insights into the reasoning behavior of modern models.

**Questions:**

1. It would be important to understand how much training on the same game set influences performance.
   For instance, the paper notes that the Duobao model was trained on classical games such as Sudoku.
   **Question:** Is it fair to compare Duobao's performance on these games with models that have *not* seen such games during pretraining?

2. In the *Impact of RL* section, could the authors clarify:
   - What fraction of KORGym games (if any) overlaps with Duobao's RL training set?
   - Whether there is an ablation isolating the RL impact from other improvements in the "thinking" model variant?

**Ethical Concerns:**

["NO or VERY MINOR ethics concerns only"]

**Limitations:**

Yes

**Quality:**

4

**Strengths And Weaknesses:**

**Strengths:**

- The proposed benchmark is highly relevant and timely for evaluating current SOTA LLMs. Unlike many existing benchmarks, KORGym is less susceptible to data contamination and better suited to test strategic thinking and planning.

- The evaluation section is particularly strong. It includes:

-- Analysis of reasoning facets in LLMs

-- Insights into the interplay between textual and visual reasoning

-- Clustering of models based on performance profiles

-- Detailed ablation studies and thinking pattern analysis

**Weaknesses:**

- The section titled “Impact of Reinforcement Learning” lacks clarity:

-- It is unclear how much overlap exists between the KORGym task set and the training tasks used for the Duobao RL training.

-- Without a well-controlled ablation, it's difficult to conclude whether the performance improvement from the non-thinking to thinking variant of the model is due to RL training alone, or to other architectural or training changes.

---

> ### Author Rebuttal · Authors · 2025-07-28
>
> First and foremost, we sincerely thank you for your recognition of KORGYM and for your valuable feedback on the analysis of game overlap and RL impact. We deeply apologize for our oversight in this regard. To quantitatively analyze the influence of game overlap and RL impact, we formulated the following formal definitions and conducted two sets of ablation experiments:
>
> **Response to Question 1 and Question 2-1**
>
> (1) **Question 1**:
>
> - Is it fair to compare Duobao's performance on these games with models that have not seen such games during pretraining?
> - What fraction of KORGym games (if any) overlaps with Duobao's RL training set?
>
> (1) **Answer 1**:
>
> 	In Doubao's technical report, only examples of Sudoku, Maze, and 21-Point were provided, without detailed data being included. Therefore, to address these two questions, we need to introduce a parameter capable of quantifying the impact of data leakage on model performance and address the problem through analytical experiments. Although we formally defined the concept of KOR in Appendix C of our submitted paper, it did not permit empirical quantification of pre-trained knowledge leakage. To rectify this, we extend the formal definition and introduce the Knowledge Leakage Coefficient, which can be empirically calculated through experiments, to further substantiate the validity of KORGYM.
>
> ---
>
> **1.1 Notational Definitions**
>
> For a game within KORGYM, the required reasoning information comprises:
>
> - **$K$**: General background or domain‑specific knowledge acquired during pre‑training and post-training.
> - **$T$**: Single‑turn or multi‑turn tasks that must be completed during gameplay.
> - **$R$**: Core game‑rule information specifically designed for solving task $T$.
> - **$A$**: The set of actions generated by the LLM to solve task $T$.
> - **$\rightarrow$**: The reasoning and interaction process whereby the LLM gradually derives the action set $A$ from the game task $T$.
> - **$S$**: The final score obtained by the LLM's interaction with the game environment based on reasoning from $R$ and/or $K$:
>   - $S(T \rightarrow A \mid K)$: Score from actions derived solely from pre‑trained knowledge $K$.
>   - $S(T \rightarrow A \mid R, K)$: Score from actions derived using both game‑rule knowledge $R$ and pre‑trained knowledge $K$.
>
> ---
>
> **1.2 Analysis of Game Redundancy**
>
> 1. **Knowledge‑Rule Decoupling**
>    Ideally, the game rule information $R$ is logically self‑contained and independent of pre‑trained knowledge $K$:
>    $$
>    R \perp K
>    $$
>
> 2. **Knowledge Leakage Coefficient**
>    The Leakage between $K$ and $R$ is quantified by the Knowledge Impact Coefficient $\beta$:
>    $$
>    \beta
>    = 1 - \frac{S(T \rightarrow A \mid R, K) - S(T \rightarrow A \mid K)}{S(T \rightarrow A \mid R, K)}
>    = \frac{S(T \rightarrow A \mid K)}{S(T \rightarrow A \mid R, K)},
>    \quad \beta \in [0,1].
>    $$
> - A smaller $\beta$ indicates less overlap between the LLM’s pre-trained knowledge and the game rules. Consequently, the LLM cannot achieve optimal scores without explicitly relying on the game rules provided in the prompt.
> - Conversely, a larger $\beta$ indicates significant overlap between the LLM’s pre-trained knowledge and the game rules, allowing the LLM to complete the game task and achieve optimal scores even without explicit game rules provided in the prompt.
>
> ---
>
> **1.3 Quantitative Experiment on Knowledge Leakage**
>
> Based on the above definition, we select a subset of games from KORGYM, including the games mentioned in Doubao's technical report(Sudoku and Maze), some classic games (Snake, Sokoban, and 8-puzzle) and original/adapted games (Pipe Game, Long Cat, ...). We conduct an ablation study in which the core game‑rule information (R) was removed and then compute the Knowledge Impact Coefficient $\beta$. The results are presented below.
>
>
> |    Model    |  Snake  | Pipe Game | Long Cat | Sokoban | Word Transformation | 8‑puzzle | Play Lines | Black White Copy | Sudoku | Maze|
> |:-----------:|:-------:|:---------:|:--------:|:--------:|:-------------------:|:--------:|:----------:|:----------------:|:----------------:|:----------------:|
> | Doubao-1-5-thinking-pro |  0.055  |   0.085   |    0     |  0.904  |         0           |  0.176   |     0      |      0.166       | 0.766 | 0.95 |
> | DeepSeek-R1             |  0.333  |   0.200   |   0.208  |  0.636  |         0           |  0.307   |     0      |      0.133       | 0.666 | 0.966 |
> | Qwen3-32B-thinking      |  0.071  |   0.080   |    0     |  0.909  |         0           |  0.321   |     0      |      0.117       | 0.888 | 0.729 |
> | o3‑mini                 |  0.956  |   0.000   |    0     |  0.948  |         0           |  0.617   |     0      |      0.166       | 0.714 | 0.976 |
> | Claude-3.7-thinking     |  0.441  |   0.200   |   0.312  |  0.461  |         0           |  0.521   |     0      |      0.222       | 1 | 1.05 |
> | Average | 0.371 | 0.113 | 0.104 | 0.771 | 0 | 0.388 | 0 | 0.160 | 0.806 | 0.934 |
>
> >From the above results, we can draw the following preliminary conclusions:
>
> > - For **Question 1**, we observe that:
> 	-  For games explicitly **trained on by Doubao-1-5-thinking-pro** (Sudoku and Maze), the β values are indeed higher but remain at a moderate level among the sampled models, which is within an acceptable range;
> 	-  For **classic games** (Snake and 8-Puzzle), while Doubao-1-5-thinking-pro demonstrates **strong performance** (ranking 3rd on Snake and 4th on 8-Puzzle among all models in the main experiments), its β values remain the **lowest** among the sampled models. This observation supports the view that reinforcement learning (RL) processes genuinely enhance a model’s reasoning capabilities rather than simply enabling it to memorize training questions or regurgitate pretraining knowledge, thus demonstrating **strong generalization**.
> 	-  For games **originally developed or adapted** by our engineering team, all models exhibit very low β values, and given that at least **80%** of KORGYM games are original or adapted, this further validates the effectiveness of KORGYM.
>
> > - For **Question 2-1**, we confirm that the original/adapted games (80%+) enable Doubao-1-5-thinking-pro to maintain relatively low β values. Even for classic games, the β values remain at a low level. Hence, we estimate that the overlap between games and Doubao’s RL training set does **not exceed 15%**. In future work, we will leverage this methodology for further experiments to determine a more precise overlap ratio.
>
> > - From a **methodological perspective**, we believe that this Knowledge Leakage Coefficient evaluation framework can effectively reveal the proportion of data types used in the LLM training process. For instance, although the training data composition is not explicitly disclosed in the reports, our experimental results suggest that **o3-mini likely involved Snake-related data and Doubao-1-5-thinking-pro likely involved Sokoban-related data**. Moving forward, we plan to further develop this methodology to uncover additional interesting insights.
>
> ---
>
> **Response to Question 2-2**
>
> (2) **Question 2**: Whether there is an ablation isolating the RL impact from other improvements in the "thinking" model variant?
>
> (2) **Answer 2**:
>
> We sincerely apologize for the oversight regarding the ablation analysis on RL impact from other improvements in the "thinking" model variant. Here, we provide additional supplementary analysis. To isolate the effect of RL training from other structural or training environment factors contributing to the performance difference from the non-thinking to the thinking variants of the model, we selected three LLMs that offer both non-thinking and thinking modes. For Gemini-2.0-Flash and Claude-3.7, the results under both modes are already reported in Table 1 and Figure 5 of our paper. The final experimental results are as follows:
>
>
>
> |    Model    |  Snake  |  Long Cat |  Black White Copy |Emoji Connect| Word Transformation |
> |:-----------:|:-------:|:---------:|:--------:|:--------:|:-------------------:|
> | Qwen3-32B-nonthinking    |  14.6  |   0.42   |   0.32  |  0.32  |         0.34           |
> | Qwen3-32B-thinking |  13.9(-0.7)  |   0.62(+0.2)    |    0.34 (+0.02)    |  0.48 (+0.16) |         0.48 (+0.14)          |
> | Claude3.7-nonthinking         |  6.9  |  0.14   |    0.02     |  0.16  |        0.22           |
> | Claude3.7-thinking      |  9.75 (+2.85) |   0.32 (+0.18)  |    0.18 (+0.16)    |  0.76 (+0.6)  |         0.56  (+0.34)        |
> | Gemini-2.0-Flash-nonthinking     |  2.05  |   0.1   |  0.02  |  0.12  |         0.12           |
> | Gemini-2.0-Flash-thinking     |  1.85 (-0.2) |   0.08 (-0.02)  |  0.16(+0.14)  |  0.18 (+0.06) |         0.18   (+0.06)        |
>
> From these results, we observe that the **thinking variants of the same model consistently outperform their non-thinking counterparts**, which supports the **validity of the performance improvements attributed to the transition from the non-thinking to the thinking variant of the model**.
>
> ---
>
> Finally, we sincerely thank you for your valuable feedback on KORGYM again, and we look forward to further discussions with you.

---

### Official Review · Reviewer_npDs · 2025-07-06

**Clarity:** 3
**Significance:** 3
**Originality:** 3
**Rating:** 5
**Confidence:** 4

**Summary:**

This paper introduces KORGym, a game-based LLMs and VLMs evaluation benchmark. The authors design 56 games across 6 reasoning dimensions and supports multi-turn, interactive assessments. The authors conduct large-scale evaluation on 19 LLMs (including 11 thinking models) and 8 VLMs. The results demonstrate KORGym’s utility for evaluating and challenging LLMs on knowledge-orthogonal reasoning abilities.

**Questions:**

Check the weakness part mentioned above

**Ethical Concerns:**

["NO or VERY MINOR ethics concerns only"]

**Final Justification:**

I am satified with the author's repsponse. My final rating is accept.

**Limitations:**

Check the weakness part mentioned above

**Quality:**

3

**Strengths And Weaknesses:**

Strengths:
Novel Benchmark: KORGym offers a diverse set of games to evaluate the reasoning abilities of LLMs and VLMs.
Broad Scope: KORGym introduces a large suite of games spanning six key reasoning dimensions. Also, the authors design evaluation method beyond binary scoring.
Ready-to-Use Implementation: The authors not only provide the evaluation datasets and metrics but also deliver optimized evaluation code. This significantly lowers the barrier for adoption and ensures reproducibility.
Strong Empirical Study: The experiments involve a broad collection of current LLMs and VLMs, with both closed- and open-source models. This enables comprehensive comparison and drives key findings on strengths, weaknesses and behavioral tendencies between model series.

Weaknesses:
Potential Domain and Task Distribution Bias: Although the knowledge-orthogonality concept aims to decouple evaluation from prior knowledge, the game/task design process is not fully transparent regarding avoidance of distributional overlap with LLM pre-training corpora. The paper does not concretely quantify the degree of knowledge leakage, which may influence some of the comparative results or limit conclusions about pure reasoning.

---

> ### Author Rebuttal · Authors · 2025-07-28
>
> Thank you for your insightful suggestions. We apologize for not clearly addressing this aspect in our KORGYM submission. Below, we will elaborate on the measures taken in the design of KORGYM to minimize overlap with pre-training data. In addition, building on the formal definition of Knowledge Orthogonal Reasoning (KOR) in our paper, we will introduce our quantifiable metrics to assess knowledge leakage and select exemplar games within KORGYM to empirically analyze the extent of such leakage.
>
> **Response to Weakness**
>
> (1) **Question1**:  Potential Domain and Task Distribution Bias: Although the knowledge-orthogonality concept aims to decouple evaluation from prior knowledge, the game/task design process is not fully transparent regarding avoidance of distributional overlap with LLM pre-training corpora.
>
> (1) **Answer1**:
>
> In the design phase of KORGYM, significant efforts were made to minimize potential knowledge leakage:
>
> > -  Unlike traditional reasoning benchmarks, KORGYM is designed as **dynamically generated games**. The nearly infinite decision space and symbolic diversity significantly reduce the likelihood of overlap with pre-training data.
> > -  Games, especially multi-turn scenarios, require **long-term strategic planning**, which is typically underrepresented in pre-training corpora.
> > -  During the selection and creation of games, we developed numerous **original games and modified traditional games** extensively to further prevent overlap with pre-training data.
>
> ---
>
> (2) **Question2**:  The paper does not concretely quantify the degree of knowledge leakage, which may influence some of the comparative results or limit conclusions about pure reasoning.
>
> (2) **Answer2**：
>
> Thank you for your valuable suggestion and we will provide a formal definition of the degree of knowledge leakage and supplement it with quantitative experiments:
>
> **2.1 Formal Definition of the Knowledge Leakage in KORGYM**
>
> In Appendix C of our submitted paper, we formally defined the concept of "Knowledge Orthogonal Reasoning" (KOR). However, it does not allow for empirical quantification of the extent of pre-trained knowledge leakage. This limitation arises from the inherent impossibility of fully eliminating the influence of pre-trained knowledge $K$ on the reasoning process. Additionally, the modeling scenario of KORBench[1], structured as a question-and-answer format, differs significantly from that of KORGYM. Consequently, we extend the formal definition originally proposed in KORBench and present an experimental approach to quantitatively measure the degree of pre-trained knowledge leakage.
>
> [1] Ma et al. KOR-Bench: Benchmarking Language Models on Knowledge-Orthogonal Reasoning Tasks. arXiv.2410.06526.
>
> ---
>
> **2.2 Notational Definitions**
>
> For a game within KORGYM, the required reasoning information comprises:
>
> - **$K$**: General background or domain‑specific knowledge acquired during pre‑training and post-training.
> - **$T$**: Single‑turn or multi‑turn tasks that must be completed during gameplay.
> - **$R$**: Core game‑rule information specifically designed for solving task $T$.
> - **$A$**: The set of actions generated by the LLM to solve task $T$.
> - **$\rightarrow$**: The reasoning and interaction process whereby the LLM gradually derives the action set $A$ from the game task $T$.
> - **$S$**: The final score obtained by the LLM's interaction with the game environment based on reasoning from $R$ and/or $K$:
>   - $S(T \rightarrow A \mid K)$: Score from actions derived solely from pre‑trained knowledge $K$.
>   - $S(T \rightarrow A \mid R, K)$: Score from actions derived using both game‑rule knowledge $R$ and pre‑trained knowledge $K$.
>
> ---
>
> **2.3 Analysis of Game Redundancy**
>
> 1. **Knowledge‑Rule Decoupling**
>    Ideally, the game rule information $R$ is logically self‑contained and independent of pre‑trained knowledge $K$:
>    $$
>    R \perp K
>    $$
>
> 2. **Knowledge Leakage Coefficient**
>    The Leakage between $K$ and $R$ is quantified by the Knowledge Impact Coefficient $\beta$:
>    $$
>    \beta
>    = 1 - \frac{S(T \rightarrow A \mid R, K) - S(T \rightarrow A \mid K)}{S(T \rightarrow A \mid R, K)}
>    = \frac{S(T \rightarrow A \mid K)}{S(T \rightarrow A \mid R, K)},
>    \quad \beta \in [0,1].
>    $$
> - A smaller $\beta$ indicates less overlap between the LLM’s pre-trained knowledge and the game rules. Consequently, the LLM cannot achieve optimal scores without explicitly relying on the game rules provided in the prompt.
> - Conversely, a larger $\beta$ indicates significant overlap between the LLM’s pre-trained knowledge and the game rules, allowing the LLM to complete the game task and achieve optimal scores even without explicit game rules provided in the prompt.
>
> ---
>
> **2.4 Quantitative Experiment on Knowledge Leakage**
>
> Based on the above definition, we select a subset of games from KORGYM, including some classic games (Snake, Sokoban, and 8-puzzle) and original/adapted games (Pipe Game, Long Cat, ...). We conduct an ablation study in which the core game‑rule information (R) was removed and then compute the Knowledge Impact Coefficient $\beta$. The results are presented below.
>
>
> |    Model    |  Snake  | Pipe Game | Long Cat | Sokoban | Word Transformation | 8‑puzzle | Play Lines | Black White Copy |
> |:-----------:|:-------:|:---------:|:--------:|:--------:|:-------------------:|:--------:|:----------:|:----------------:|
> | Doubao-1-5-thinking-pro |  0.055  |   0.085   |    0     |  0.904  |         0           |  0.176   |     0      |      0.166       |
> | DeepSeek-R1             |  0.333  |   0.200   |   0.208  |  0.636  |         0           |  0.307   |     0      |      0.133       |
> | Qwen3-32B-thinking      |  0.071  |   0.080   |    0     |  0.909  |         0           |  0.321   |     0      |      0.117       |
> | o3‑mini                 |  0.956  |   0.000   |    0     |  0.948  |         0           |  0.617   |     0      |      0.166       |
> | Claude-3.7-thinking     |  0.441  |   0.200   |   0.312  |  0.461  |         0           |  0.521   |     0      |      0.222       |
> | Average | 0.371 | 0.113 | 0.104 | 0.771 | 0 | 0.388 | 0 | 0.160 |
>
>
>
> From the above results, we can draw the following preliminary conclusions:
>
> > - **Model perspective:** Models such as **o3‑mini** and **DeepSeek‑R1** exhibit relatively notable knowledge leakage, whereas other models (e.g., **Doubao‑1‑5‑thinking‑pro**) demonstrate comparatively lower leakage. This observation indirectly supports the view that **reinforcement learning (RL) processes genuinely enhance a model's reasoning capabilities, rather than merely enabling it to memorize training questions or regurgitate knowledge from pretraining**.
> > -  **Game perspective:** Knowledge leakage is more prominent in **classic games**, while the corresponding β values for **original/adapted games** remain at a relatively low level. Given that more than **80%** of the games in KORGYM are originally developed or adapted by our engineering team, this strongly supports the validity and reliability of KORGYM.
> > -  **Methodological perspective:** We believe that the knowledge leakage quantification experiment itself offers an interesting insight and methodology for **analyzing models’ genuine reasoning abilities and uncovering information contained within the pre-training data**. Future work could extend this approach to conduct deeper analyses and evaluations of reasoning capabilities.
>
> Once again, we sincerely thank you for your valuable feedback on KORGYM. Your comments have provided insightful inspiration and an important reminder. Moving forward, we plan to further increase the proportion of original/adapted games in KORGYM to more accurately capture models' genuine reasoning capabilities.

---

> > ### Comment · Reviewer_npDs · 2025-08-05
> >
> > I am satified with the author's response. My final rating is accept.

---

> > > ### Author Response · Authors · 2025-08-06
> > > **Thank you for your valuable feedback!**
> > >
> > > Thank you very much for your thoughtful feedback and for continuing to support the acceptance of our work. We sincerely appreciate your recognition and support—your encouraging feedback is valuable to us and greatly motivates our ongoing research efforts.

---

### Note · Authors · 2025-08-13

We sincerely thank all reviewers and the AC for their constructive feedback, which has greatly strengthened our work. Throughout the rebuttal and discussion period, we carefully addressed each concern with additional experiments, formal definitions, and quantitative analyses to provide clear evidence supporting our claims.

In particular:

* **Knowledge Orthogonality Validation** – We extended our formal definition and introduced the *Knowledge Leakage Coefficient*, enabling empirical quantification of overlap between pre-trained knowledge and KORGym tasks. Experiments across multiple LLMs confirmed that more than 80% of KORGym's original/adapted games exhibit minimal leakage, supporting its validity and reliability. (See Reviewer npDs.Q2, Reviewer V7Eo.Q1, Reviewer SShB.Q1)
* **RL Impact and Game Overlap** – We conducted ablations isolating RL effects from other architectural changes, and estimated that the overlap between Doubao's RL training set and KORGym games does not exceed 15%. These results indicate that RL contributes genuine reasoning improvements rather than memorization. (See Reviewer V7Eo.Q1, Q2)
* **Correlation with Specialized Benchmarks** – Spearman correlation analyses with domain-specific benchmarks (e.g., MathArena, MC Bench, OmniSpatial) showed strong alignment, demonstrating that KORGym meaningfully reflects reasoning ability across diverse tasks.(See Reviewer SShB.Q3)
* **Related Work and Presentation** – We integrated key references and will further refine visual figures and explanations in the final version.(See Reviewer 3dhT.Q1, Q2)

We are grateful that reviewers acknowledged these improvements and, in most cases, raised their scores. We believe KORGym is now a robust, reproducible, and timely contribution, offering a dynamic, game-based, RL-supported, multi-turn evaluation framework that addresses critical gaps in LLM reasoning assessment. We look forward to seeing it serve as a valuable resource for both researchers and practitioners in advancing the understanding and development of reasoning-capable LLM systems.

---

### Decision · Program_Chairs · 2025-09-17

**Decision:**

Accept (spotlight)

**Comment:**

This paper presents KORGym, a dynamic, game-based benchmark for evaluating reasoning in LLMs and VLMs. The benchmark is well-motivated, offering over 50 interactive tasks across diverse reasoning dimensions, with support for multi-turn and RL settings. Reviewers highlighted its novelty, breadth, and strong empirical evaluation across many models, noting that it fills an important gap in assessing reasoning beyond domain-specific tests. I recommend acceptance.